EMBO
Molecular Medicine

# Blocking microglial reactivity via purinergic receptors prevents subacute cognitive deficits after TIA

Gemma Llovera [1], Steffanie Heindl [1], Daniel P Varga[1], Nikolett Lenart[2], Sebastian Kallabis [3,4], Vanessa Göb[5], David Stegner [5], Raphael Escaig[6], Leo Nicolai[6,7], Nicolai Franzmeier [1,8,9], Felix Meissner [3], Adam Denes [2,10] & Arthur Liesz [1,8✉]

## Abstract

Our research presents a new animal model of transient ischemic attack (TIA) that mimics brief episodes without cell loss, but results in neuronal and behavioral deficits. We identified excessive microglial reactivity, driven by acute ATP release, as a key factor in post-TIA neurological deficits, which were ameliorated by inhibiting the P2Y12 receptor, a microglia-specific purinergic receptor in the brain parenchyma responsible for activity-dependent microglial cell-cell interactions. This finding suggests that modulation of microglial reactivity offers a promising strategy to prevent cognitive impairment in TIA patients, opening avenues for future research in this underexplored area.

**Keywords** ATP; Microglia; TIA; Purinergic Receptor; Synapse
**Subject Category** Neuroscience

See also: P Zheng et al

## Introduction

Transient ischemic attacks (TIAs) are a prevalent form of cerebrovascular disease in which rapid reperfusion of an occluded cerebral artery instantly restores blood flow to the affected region, resulting in a transient neurological deficit without detectable parenchymal injury (Easton et al, 2009). As such, TIA differs mainly from ischemic or hemorrhagic stroke, the other most common cerebrovascular disorders, in that it does not cause detectable structural damage to brain tissue. Nevertheless, TIAs are associated with short- and long-term risks, such as cardiovascular incidents, cognitive impairment, and increased mortality (Giles and Rothwell, 2007; Hill et al, 2004; Navis et al, 2019; Sivakumar et al,

2014; Touze et al, 2005; Wu et al, 2007). Most notably, TIA patients have a similarly increased risk of developing dementia as those who have had a stroke despite the lack of structural tissue damage after TIA (Luengo-Fernandez et al, 2013; Pendlebury et al, 2019).

Annually, around 500,000 TIA cases are reported in the United States alone and up to 60 cases per 100,000 across Western countries (Brown et al, 1998; Cancelli et al, 2011; Degan et al, 2017), marking TIA as an early warning sign critical for preventing stroke and dementia. However, current therapies for TIA are limited to control of cardiovascular risk factors (Johnston et al, 2018; Powers et al, 2019; Wang et al, 2013), while pathomechanisms of TIA and their role in post-TIA cognitive deficits remain elusive and so far unexplored.

## Results and discussion

In the past decade, the clinical diagnosis of TIA has faced a paradigm shift from "time-based" (focal deficits for less than 1 h) to a "tissue-based" definition (acute neurological symptoms without brain infarction) (Degan et al, 2017; Hurford et al, 2019). Some previous work have described different mouse model of TIA (Pedrono et al, 2010; Quenault et al, 2017; Wang et al, 2020), but its characterization, in our view, was not sufficient. Consequently, given our experience with murine stroke models, we took the classic model of intraluminal middle cerebral artery (MCA) occlusion (Llovera et al, 2021) as a reference, and adapted it to establish a model of TIA and characterized it in more detail. A brief five-minute occlusion of the MCA was sufficient to induce significant behavioral deficits both displaying general and focal deficits both in male and females (Figs. 1A and EV1A,B) as well as impaired spatial memory up to two days after TIA in males (Fig. 1B). However, these brief ischemic episodes leading to transient tissue hypoxia (Fig. EV1C) did not induce any detectable cell death, change in blood–brain barrier permeability, neuronal loss, changes in glucose metabolism by in vivo positron-emission tomography or up-regulation of cytoskeleton proteins in the extracellular space,

[1]Institute for Stroke and Dementia Research (ISD), LMU University Hospital, LMU Munich, Munich, Germany. [2]Laboratory of Neuroimmunology, HUN-REN Institute of Experimental Medicine, Budapest, Hungary. [3]Department of Systems Immunology and Proteomics, Institute of Innate Immunity, University Hospital Bonn, Bonn, Germany. [4]Core Facility Translational Proteomics, Institute of innate Immunity, University Hospital Bonn, Bonn, Germany. [5]Julius-Maximilians University of Würzburg, Rudolf Virchow Center for Integrative and Translational Bioimaging and University Hospital Würzburg, Würzburg, Germany. [6]Medizinische Klinik und Poliklinik I, LMU University Hospital, Munich, Germany. [7]DZHK (German Centre for Cardiovascular Research), Partner Site Munich Heart Alliance, Munich, Germany. [8]Munich Cluster for Systems Neurology (SyNergy), Munich, Germany. [9]University of Gothenburg, The Sahlgrenska Academy, Institute of Neuroscience and Physiology, Department of Psychiatry and Neurochemistry, Mölndal and Gothenburg, Gothenburg, Sweden. [10]Mercator Fellow, Institute for Stroke and Dementia Research, University Hospital, LMU Munich, Munich, Germany. ✉E-mail: Arthur.Liesz@med.uni-muenchen.de

confirming the absence of any structural tissue damage (Fig. EV1D–H). We also performed cerebral hypoperfusion for 5 min using a bilateral common carotid artery occlusion (BLCCAO) model, which was not sufficient to induce behavioral changes, microglia activation or a decrease in the number of synapses (Fig. EV2A–C). Therefore, this model closely resembles the pathophysiological definition of TIA (Perry et al, 2022) and allows in-depth functional studies.

After observing neurological deficits, we set out to explore the pathophysiological mechanisms mediating prolonged functional deficits after TIA by performing repetitive wide-field neuronal calcium imaging to analyze cortical network function (Cramer et al, 2019). To our surprise, we found that a transient ischemic stroke led to a significant reduction in global interhemispheric network connectivity for more than 14 days, driven by a specific lack of connectivity in the network of the somatosensory cortex supplied by the MCA, whereas other cortical areas not supplied by the affected MCA did not suffer significant alterations (Fig. 1C,D). These results suggest a very prolonged and specific effect of the brief TIA period on cortical network function in the affected vascular brain area (somatosensory cortex). As we found stable neuronal survival (Fig. EV1F), we analyzed subcellular remodeling using spine density by histology, noting a significant reduction in dendritic spine density 24 h after TIA (Fig. 1E). Correspondingly, glutamatergic synapse density in the somatosensory cortical area of the MCA territory was significantly reduced (Fig. 1F).

Age is a crucial factor in transient ischemic attacks (TIAs) because it significantly increases both the risk of having a TIA and the likelihood of developing other health problems that contribute to vascular events (Khare, 2016). As people age, they often accumulate diseases such as hypertension, diabetes, and atrial fibrillation, which increase the risk of TIA (Ippen et al, 2021). Consequently, older people who suffer a TIA tend to have worse outcomes, such as higher rates of subsequent stroke, rehospitalization, and shorter life expectancy (Fasth et al, 2021; Gattellari et al, 2012). Recognizing age as a key comorbidity is vital for effective risk assessment and personalized management of TIA patients. For this reason, we decided to test the effects of TIA in older animals to see if we could observe an exacerbated effect of the TIA phenotype. We observed a similar but more pronounced effect in older animals (20-month-old), both in their Neuroscores and in the reduction of synapses; however, this effect remained transient (Fig. EV3A–C). Although cohorts cannot be directly compared, we observed a reduction in glutamatergic synapses in sham-aged animals compared to sham-young animals (compare Figs. EV3B and 1F), indicating that aging alone already has an effect on synaptic density.

Since the underlying cause of the observed TIA phenotype was unclear, we set out to look for changes in the extracellular compartment to better understand the possible mechanisms involved. Using microfluidic perfusion of the extracellular space (Birngruber et al, 2013), in the somatosensory cortex, we sampled up to 14 d after TIA. These perfusates were used for proteomic analyses by liquid chromatography coupled with mass spectrometry (LC-MS). We identified a specific protein cluster regulated in the acute to subacute phase (cluster 3) and a group of secreted proteins upregulated in the chronic phase (cluster 5) (Fig. 2A,B). Closer examination of groups 3 and 5 revealed a peak of protein intensity at 24 h in group 3, while group 5 showed a steady increase in protein intensity expression over time (Fig. 2C,D), reinforcing our previous observation (Fig. 2A,B). Further analysis of the

pathways of the ten most upregulated genes in each of these two groups indicated a large number of genes related to stress and inflammatory response, indicating a chronic inflammatory response after TIA (Fig. 2C,D). The key limitation is that the proteomics analysis of the microfluidic perfusate was biased towards highly abundant proteins (such as albumin and hemoglobin), which dominate the sample's protein content (the top 20 proteins constitute almost 50% of the total abundance). This high dynamic range in protein concentration, combined with the limited sample volume, significantly reduced the sensitivity for detecting low-abundant, yet biologically critical, proteins like cytokines, potentially leading to an incomplete or biased picture of the extracellular protein environment. Future studies would benefit from using techniques, such as the Seer Proteograph technology, to circumvent this high dynamic range challenge.

To identify which cell populations in the cortex were most transcriptomically affected after TIA, we performed single-cell RNA sequencing of cells isolated from cortical tissue homogenates in control animals and at 24 h and 3 d after TIA. Surprisingly, neurons showed only a very minor transcriptional regulation, while the largest number of differentially regulated genes was detected in the microglia population, particularly at 3 d after TIA (Fig. 2E,F). Further analysis of genes differentially regulated in the microglial cell population revealed S100a, Ddit4 and Cd300lf being among the most highly upregulated genes at 3 d after TIA. These genes are pivotal in responding to cellular stress, including hypoxia and neuroinflammation (Denstaedt et al, 2018; Perez-Sisques et al, 2021; Voss et al, 2015) (Fig. 2G), which was also reflected by the corresponding pathway analysis of regulated genes in the microglia cell cluster (Fig. EV4A,B).

Given that microglial cells showed the most significant transcriptional changes, we focused on studying these cells in more detail. To do so, we performed a specific analysis of microglia using Nanostring technology, examining selected microglial populations up to 7 days after TIA, which confirmed long-lasting transcriptional changes with proinflammatory polarization in microglial reactivity still at 7 d post-TIA (Fig. EV4C). Based on these transcriptomic findings indicating an important role of microglia in the post-TIA brain, we further investigated the cellular function and morphology of microglia. Structural analysis of microglial cells in the somatosensory cortex, by automated 3D-morphometry (Heindl et al, 2018), identified a significant change in microglial morphology only 24 h after TIA, which suggests a transient reactive phenotype in males (Fig. EV5A) and also in females (Fig. EV5B). Correspondingly, in vivo analysis of microglia confirmed a significant increase in process motility (Fig. EV5C)—a microglial phenotype that has been previously associated with altered cell–cell interaction and potential neurotoxicity (Block et al, 2007; Cserep et al, 2020). Activation of microglia after a TIA can also be observed in older animals (20 months old), but here the microglial cells remained activated for up to 7 days after the TIA (Fig. EV3C).

Since we detected neuronal dysfunction (Fig. 1) as well as microglial reactivity in both histology and transcriptome (Figs. EV4 and 5), we decided to investigate the two main functions of microglial cells already described in the literature, neuronal surveillance (Cserep et al, 2020) and synapse pruning (Hong et al, 2016). Therefore, we further investigated the microglia–neuron interaction as a potential basis of prolonged neuronal dysfunction after TIA. Specifically, microglial interaction

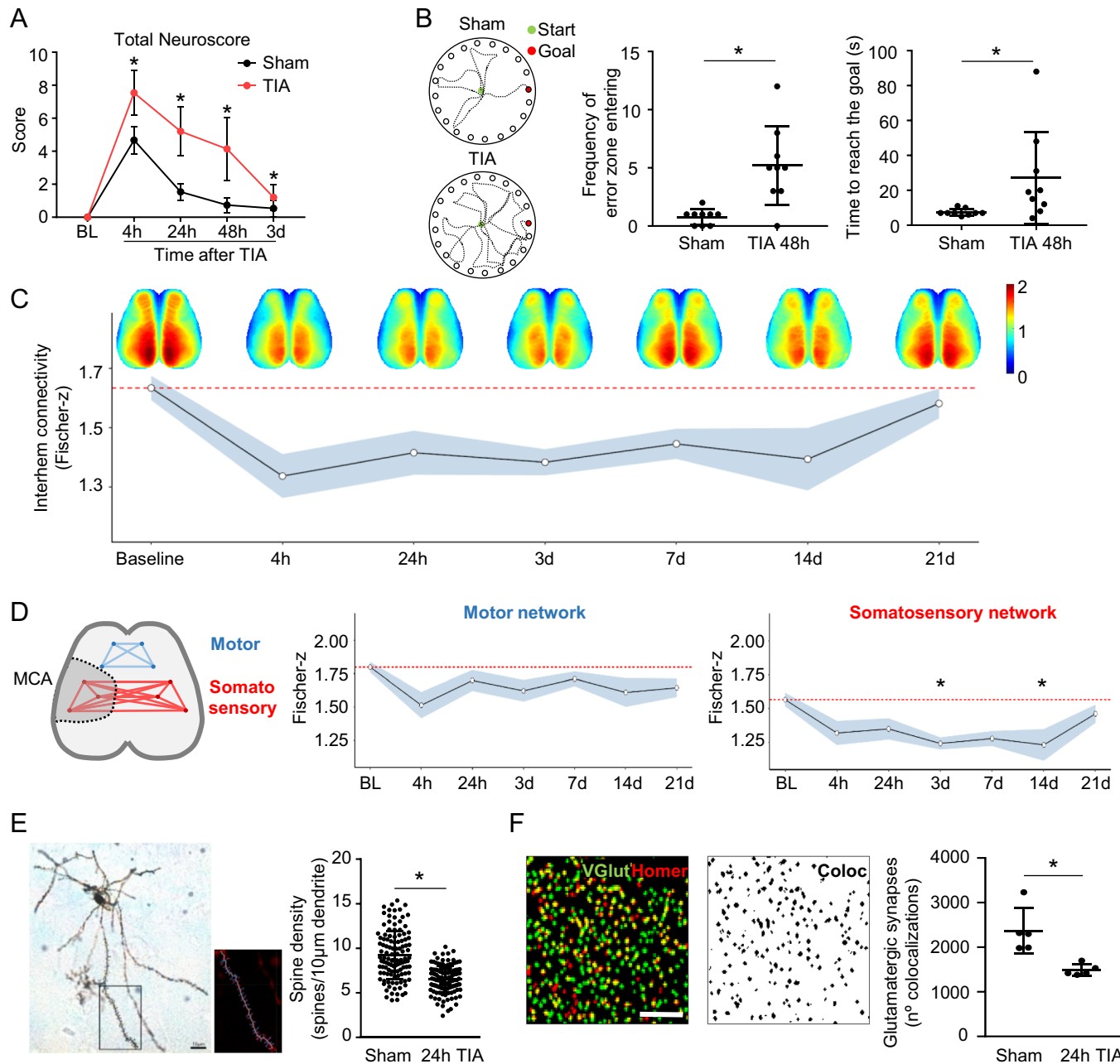

**Figure 1. Brief transient ischemia, in the absence of tissue injury, leads to prolonged neuronal dysfunction.**

(A) Neuroscore at different time points after TIA in male animals ($n = 15$ per group; 4, 24, and 48 h:$p$ value <0.0001, 3 d:$p$ value = 0.023) and (B) mean average of mice trajectories to reach the goal; frequency of errors ($p$ value = 0.0023) and time to reach the goal ($p$ value = 0.0070) in the Barnes Maze test at 48 h after TIA in males ($n = 9$ per group). (C) Global interhemispheric connectivity and (D) network connectivity for motor and somatosensory networks at indicated time points before in males (Baseline and red dotted line) and after TIA ($n = 8$ per time point; 3 d:$p$ value = 0.032 and 14 d:$p$ value = 0.044) (MCA middle cerebral artery vascular territory). (E) Representative image of Golgi-Cox stained pyramidal neuron 24 h after TIA in males, scale bar = 10 µm. Magnification show an example of 3D reconstruction for a dendrite section with spines used for their quantification ($n = 5$ per group; $p$ value <0.0001). (F) Representative particle image of presynaptic terminals by VGlut1 (green) and postsynaptic terminals by Homer1 (red), as used for quantification of colocalized presynaptic and postsynaptic particles (black) 24 h after TIA in males ($n = 5$ per group; $p$ value = 0.0079), scale bar = 10 µm. In (C, D): black line: mean and gray: Standard Error. Statistical tests: (A) two-way ANOVA, corrected for multiple comparisons using two-stage step-up method of Benjamin Kriegel. (B, E, F) T-test. (C) repeated measure of ANOVA followed by Turkey's post hoc test. Error bars indicate ±SD. *$p$ < 0.05. Source data are available online for this figure.

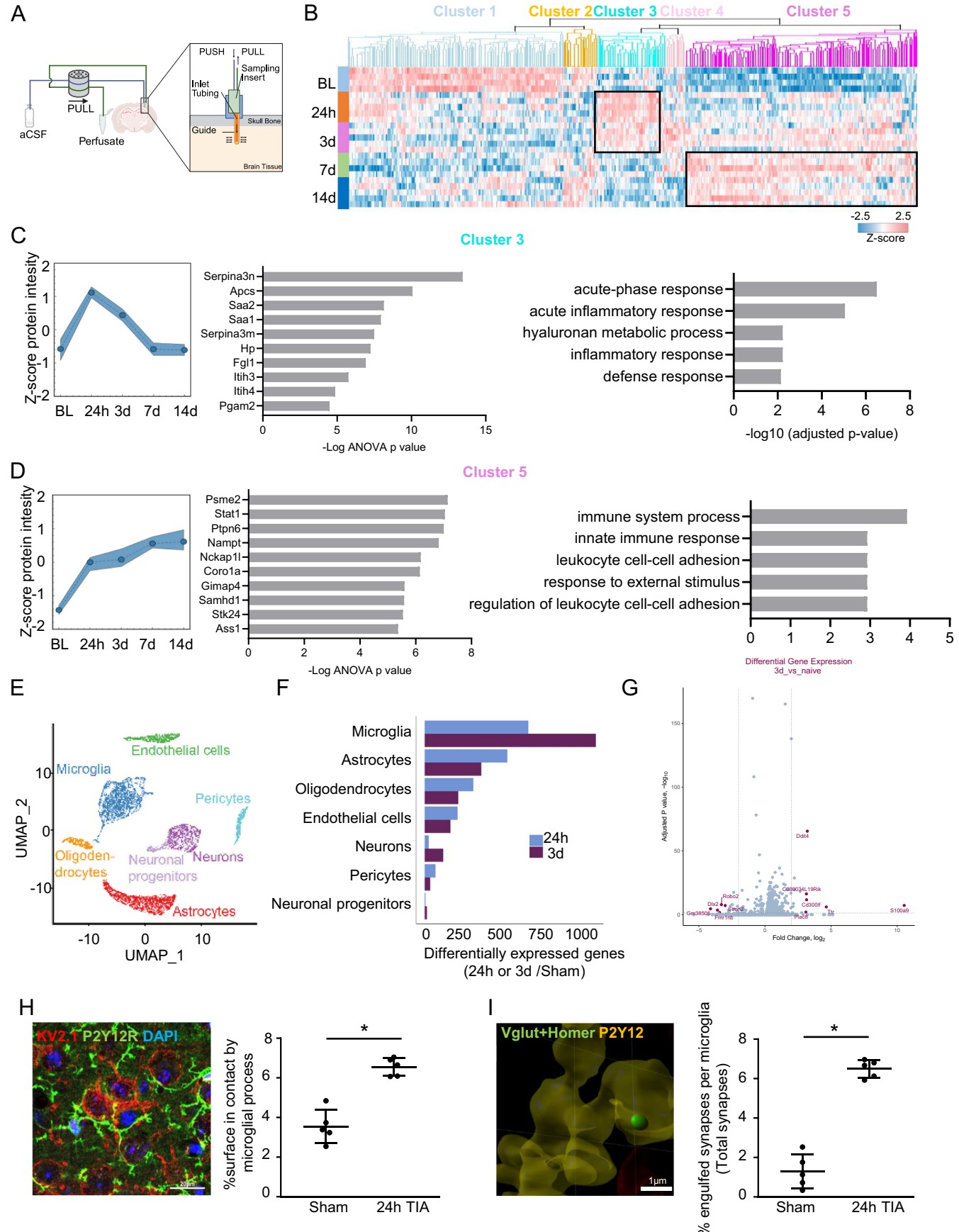

**Figure 2. TIA triggers a cortical proinflammatory environment and microglial reactivity.**

(A) Representative scheme of the microperfusion system. (B) Heat map of all ANOVA+ proteins of the LC-MS analysis of the perfusate products, (C) protein intensity over time, top ten upregulated genes and pathway analysis from cluster 3 and (D) cluster 5 in males ($n = 5$ per group). (E) Uniform manifold approximation and projection (UMAP) plot, colored by identified clusters (different cell types). (F) Number of differentially expressed genes ($p < 0.05$) at 24 h and 3 d after TIA in each cell type compared to control (naïve) in males ($n = 3$ per group). (G) Volcano plot of differentially regulated genes in microglia 3 d compared to naïve animals. Cut-off for illustrating gene names: Log2FC >|2| illustrated by vertical gray dashed lines, adjusted $p$ value: >−log10(0.05) illustrated by horizontal gray dashed line ($n = 3$ per group). (H) Representative image and quantification of microglia (P2Y12R+) and neuron soma (KV2.1+) contacts 24 h after TIA in males ($n = 5$ per group; $p$ value = 0.0079), scale bar = 20 μm. (I) Representative image of microglia (P2Y12+) and synapse (Vglut+Homer+) staining for synapse engulfed quantification 24 h after TIA in males ($n = 5$ per group; $p$ value = 0.0079), scale bar = 1 μm. Statistical tests: (B–D) Two-way ANOVA, Fisher's exact testing. (G) Wilcoxon rank sum test + Bonferroni correction. (H, I) Two-way Student's $t$-test. Error bars indicate ±SD. *$p < 0.05$. Source data are available online for this figure.

with neuronal somata (termed somatic purinergic junctions) were identified to be critical for microglial sensing of neuronal health (Cserep et al, 2022); this interaction between P2Y12+ microglia and Kv2.1+ neurons was substantially increased after TIA (Fig. 2H). Similarly, the contact between microglia and Vglut+Homer1+ excitatory synapses was also substantially increased after TIA (Fig. 2I), suggesting increased microglia–neuron interaction as a potential contributor to prolonged neuronal dysfunction after TIA.

Next, we aimed to identify the molecular mechanisms leading to the excessive microglial reactivity to the brief ischemic period during TIA. The rapid depletion of oxygen and glucose during a TIA leads to a significant increase in extracellular ATP, which acts as a crucial damage-associated molecular pattern (DAMP). When we checked for extracellular ATP concentration after TIA, we observed a rapid and massive increase within minutes after TIA which remained elevated until 24 h post-TIA (Fig. 3A). This phenomenon was confirmed by in vivo imaging using an extracellular ATP reporter system by AAV-mediated expression of the hsyn-cATPs sensor in the MCA-supplied area of Cx3cr1-tdT microglial reporter mice, which revealed an increase of focal ATP bursts after TIA (Fig. 3B). This surge in extracellular ATP was associated with rapid recruitment of microglial processes within 60-180 s after the ATP burst (Fig. 3C; Movie EV1). Remarkably, these events influence microglial function through P2Y12R and CX3CR1 pathways (Haynes et al, 2006; Huang et al, 2024). Additionally, these pathways contribute to the generation of focal ATP events, which could be highly relevant for transient ischemic attacks (TIA) and other forms of relatively subtle injury. Recent data show that microglial process recruitment to focal ATP events below 1 μM takes place via P2Y12R-mediated effects that have major effects on microglial activity and cell morphology (Berki et al, 2024). To test the causal role of increased extracellular ATP concentration for microglial reactivity and neuronal dysfunction, we injected ATP into the cisterna magna (CM) of naïve mice, which was sufficient to dose-dependently induce a similar phenotype on microglial morphology and glutamatergic synapse loss as TIA (Fig. 3D,E).

This sudden increase in ATP can originate from multiple sources: acutely stressed or damaged neurons, which release ATP into the synaptic cleft, and astrocytes, which respond with a rapid release of gliotransmitters. Most critically, this high concentration of extracellular ATP rapidly activates resident microglia through purinergic receptors, initiating a robust innate immune and inflammatory response (Rodrigues et al, 2015). While the exact cellular source of the extracellular ATP increase remains unclear, we postulate that astrocytes, neurons, and tentatively other resident brain cells contribute to its surge after TIA. In the injured brain, the main source of focal ATP events are considered to be astrocytes (Chen et al, 2024). This assumption is supported by the observation

that the deficiency in ATP-release from platelets—another potential major source of rapid ATP release from the intravascular compartment, particularly in vascular injury—does not affect the post-TIA phenotype in Unc13d-deficient mice—mice with Munc13-4-deficient platelets, which are not able to degranulate the dense granules, so there is no ATP release (Fig. EV6A–C).

ATP sensing by purinergic receptors is a well-described mechanism of microglial activation with P2Y12 as one of the most abundant and microglia-specific purinergic receptors (Cserep et al, 2020; Davalos et al, 2005; Horvath et al, 2014). Therefore, we tested the potential of a pharmacological P2Y12 receptor blockade (P2Y12Ri), PSB-0739, on preventing post-TIA microglial effects. Indeed, P2Y12Ri treatment prevented morphological changes of microglia and normalized microglia-synapse interaction frequency (Fig. 3F,G), which was associated with significantly increased numbers of glutamatergic synapses after TIA in the treatment group (Fig. 3H). Importantly, inhibition of purinergic signaling after TIA was able to significantly improve focal neurological deficits and prevent the cognitive deficits in spatial memory after TIA (Fig. 3I,J). In addition, we found no effect on platelet aggregation in the systemic circulation (Fig. EV6D,E).

This robust study established a well-characterized mouse model of transient ischemic attack (TIA) that successfully replicates the tissue-based clinical definition, demonstrating prolonged functional deficits (synapse loss and network disconnectivity) without structural damage. The core strength lies in identifying a novel pathogenic mechanism after TIA: the TIA induces a rapid increase in extracellular ATP, triggering P2Y12 receptor-mediated microglial activation and subsequent synaptic pruning, a pathology which was reversed by P2Y12 receptor inhibition. However, the precise cellular source of the critical extracellular ATP surge remains unclear. Additionally, the study is limited by the absence of an a priori power calculation; our sample size was not prospectively determined to detect a specific effect size. This approach may have implications for the statistical power of the study and suggests that the results, particularly non-significant findings, should be viewed as exploratory. The study's main contribution lies in generating new hypotheses, which can now be tested in future, adequately powered studies. This will ensure that our findings are validated and their broader significance is firmly established.

Collectively, our findings support targeting purinergic microglial reactivity as a promising immunomodulatory approach to prevent secondary neurological deterioration in patients with transient ischemic attacks. We anticipate that our findings, along with our extensively characterized animal model of TIA, will lay the groundwork for future investigations into the mechanisms and therapeutic interventions for this highly prevalent yet under-explored neurological condition.

# Methods

### Reagents and tools table

| Reagent/resource | Reference or source | Identifier or catalog number |
| --- | --- | --- |
| **Experimental models** | | |
| C57BL6/J (*M.Musculus*) | Charles river | Strain: 632 |
| C57BL/6J-Tg (Thy1-GCaMP6s) GP4.12Dkim/J (*M.Musculus*) | Dana et al, 2014 | Institute for Stroke and Dementia Research, Munich |
| CX3CR1<sup>GFP/+</sup> (*M.Musculus*) | Jaxson | Strain: 005582 |
| CX3CR1<sup>tdTomato</sup> (*M.Musculus*) | Jaxson | Prof.A. Denes at the HUN-REN Institute of Experimental Medicine in Budapest |
| Munc13-4-null (Unc13dKO) (*M.Musculus*) | Stegner et al, 2013 | Prof. D. Stegner at the Institute for Experimental Biomedicine, University of Würzburg |
| **Recombinant DNA** | | |
| **Antibodies** | | |
| Guinea pig anti-VGlut1 | Millipore | AB5905 |
| Chicken anti-Homer | Synaptic Systems | 160006 |
| rabbit anti-P2Y12 | AnaspecInc | AS-55043A |
| mouse anti-KV2.1 | NeuroMab | 75-014 |
| rabbit anti-Iba1 | Wako | 019-19741 |
| anti-NeuN antibody | MerkMillipore | MAB377 |
| AF488 goat anti-Guinea pig | Molecular Probes | A-11073 |
| AF647 goat anti-Chicken | Invitrogen | A-21449 |
| AF488 goat anti-rabbit | Thermo Fisher Scientific | A-11034 |
| AF647 goat anti-mouse | Invitrogen | A-21235 |
| AF594 goat anti-rabbit | Invitrogen | A-11012 |
| AF488 goat anti-mouse | Invitrogen | A11001 |
| DAPI (4'-6-Diamidino-2-Phenylindole-dihydrochloride) | Thermo Fisher Scientific | D3571 |
| **Oligonucleotides and other sequence-based reagents** | | |
| **Chemicals, enzymes and other reagents** | | |
| P2Y12R inhibitor, PSB-0739 | Bio-Techne Corp | 3983 |
| artificial cerebral spinal fluid (aCSF) | Bio-Techne Corp | 3525 |
| Adenosine 5-triphosphate disodium salt hydrate | Sigma | A1852-1VL |
| Ambion™ Nuclease-Free water | Invitrogen | AM9937 |
| green ATP sensor | WZ Biosciences | YL006006-AV9 |
| [18 F]FDG tracer | LMU University Hospital | N/A |
| Evans blue | Sigma-Aldrich | E2129 |

| Reagent/resource | Reference or source | Identifier or catalog number |
| --- | --- | --- |
| formamide | Roth | 6949.1 |
| Paraformaldehyde | Morphisto | Cat# 11762.00100 |
| Triton X-100 | Sigma-Aldrich | Cat# 10241957 |
| Tween 20 | Roth | Cat# 9127.1 |
| Bovine Serum Albumin | Sigma-Aldrich | Cat# 9048-46-8 |
| cold fish skin gelatine | Sigma-Aldrich | G7041 |
| TUNEL apoptosis detection kit | Millipore | S7110 |
| Hypoxyprobe-1 Green Kit | HPI | HP6-XXX |
| Donkey serum | Abcam | Cat# ab7475 |
| Goat serum | Thermo Fisher Scientific | Cat# 16210064 |
| Aldehyde fixative solution | Bioenno | 003780 |
| Impregnation solution | Bioenno | 003760 |
| DMEM | Invitrogen | Cat# 12634-010 |
| Fetal calf serum (FCS) | Gibco | Cat# 105000-064 |
| Digestion mix [DMEM + 10% FCS + 0.4%DNASEI | Roche | 11284932001 |
| CollagenaseD | Roche | 11088866001 |
| Agarose | VWR Chemicals | 9012-36-6 |
| **Software** | | |
| GraphPad Prism 7/9/10 | GraphPad Software Inc. | N/A |
| FIJI (ImageJ 2, Version 2.0.0-rc-69/1.52p & 2.14.0/1.54f) | NIH | N/A |
| Imaris (Version 8.4.0) | Bitplane | N/A |
| ZEN black edition | Zeiss | N/A |
| MATLAB (R2016b) | Mathworks | N/A |
| R version 4.0.3/4.3.1 | CRAN | N/A |
| MES | Femtonics | v.5.3560 |
| Microglia Morphology Quantification Tool (MMQT) | Heindl & Gesierich et al | N/A |
| CellRanger v.7.1.0/v.8.0.0) | 10x Genomics | N/A |
| **Other** | | |
| Silicon-coated filament | Doccol | 602112PK5Re |
| Silicon-coated filament | Doccol | 602156PK10 |
| Laser Doppler | Perimed | |
| Confocal microscope | Zeiss | Zeiss 880 |
| Spectrophotometer plate reader at 620 nm | Bio-Rad | 168-1130 |
| Barnes Maze | Noldus | |
| Quick Base | Parkell C&B metabond | S398 |
| L-Powder clear | Parkell C&B metabond | S399 |

| Reagent/resource | Reference or source | Identifier or catalog number |
|---|---|---|
| Universal Catalyst | Parkell C&B metabond | S371 |
| Artificial cerebrospinal fluid (aCSF) | https://www.alzet.com/guide-to-use/preparation-of-artificial-csf/ | |
| Guide/Dummy | Joanneum research | cOFM-GD-2-1 |
| Sampling insert | Joanneum research | cOFM-S-3 |
| Perfusate bag 10 ml | Joanneum research | OFM-BAG |
| Low-binding tubing | Joanneum research | OFM-PP2-100-LB |
| Microperfusion pump | Joanneum research | MPP102-II-PC |

## Mice

The experiments were conducted following national guidelines for the use of experimental animals, and all protocols were approved by the German governmental committees (Regierung von Oberbayern, Munich, Germany and Regierung von Unterfranken, Würzburg, Germany). Wild-type C57BL6/J mice were purchased from Charles River, C57BL/6J-Tg (Thy1-GCaMP6s) GP4.12Dkim/J (Dana et al, 2014) heterozygous mice were bred at the Institute for Stroke and Dementia Research, Munich. CX3CR1$^{GFP/+}$(JAX stock no: 005582), CX3CR1$^{tdTomato}$ (offsprings of JAX stock no: 020940 with JAX stock no: 007905) were bred by the laboratory of Prof. A. Denes at the HUN-REN Institute of Experimental Medicine in Budapest. Munc13-4-null (Unc13dKO) mice (Stegner et al, 2013) and WT-littermate controls were bred by Prof. D. Stegner at the Institute for Experimental Biomedicine, University of Würzburg. All animals used for this study were males or females at 10–12 weeks of age. One female cohort (10–12 weeks old) was used for Neuroscore and microglia morphology analysis. Another cohort of 20-month-old male mice was used for the analysis of the Neuroscore, glutamatergic synapses and microglia analysis. All animals were housed under controlled temperature (22 °C ± 2 °C), with a 12 h light-dark cycle period and access to pelleted food and water ad libitum. All data are reported according to the ARRIVE criteria.

All surgeries were performed by the same surgeon (G.L.) at the Institute for Stroke and Dementia Research (ISD) animal facility. Only the experiment shown in Fig. EV6 was performed in Würzburg (also by G.L.), since the transgenic animals were located at the Institute for Experimental Biomedicine (Prof. D. Stegner´s lab) and could not be imported to the ISD in Munich. The only exception is the experiment shown in Fig. 3B,C and Movie EV1 that was performed in Budapest (Prof. A. Denes' lab) by R.F.

## Transient ischemic attack (TIA) model

TIA was performed using a modification of the intraluminal filament technique as previously described (Llovera et al, 2021), animals were anesthetized (isoflurane in 30%$O_2$/70%$N_2O$) and received an incision between ear and eye to expose the temporal skull. A laser Doppler probe was placed over the skull above the middle cerebral artery (MCA) territory. Animals were then placed in a supine position. After a midline neck incision, the common carotid artery and left external carotid artery were isolated and ligated, a 2-mm-silicon-coated filament (Doccol, #602112PK5Re) was inserted into the internal carotid artery, through the external carotid artery, until the occlusion of the MCA, as checked by a corresponding laser Doppler flow reduction. After 5 (TIA), 15, or 30 min of MCA occlusion (MCAo), the filament was removed. For the survival period, animals were kept in their homecage with facilitated access to water and food. For 2 P imaging, a 6–0 210 um tip filament with 10-mm silicon rubber coating (#602156PK10 Doccol) was used. Sham-operated mice received the same surgical procedure, except the filament was not inserted. Predefined exclusion criterion: Mice without a reduction in blood flow to <20% of the baseline, controlled by laser Doppler flow (no animals were excluded for this criterion).

## Bilateral middle cerebral artery occlusion (BLCCAO)

Animals were anesthetized (isoflurane in 30%$O_2$/70%$N_2O$) and placed in a supine position. After a midline neck incision, the right and left common carotid arteries were isolated and clipped. After 5 min both clips were removed. For the survival period, animals were kept in their homecage with facilitated access to water and food.

## Cisterna magna injection for P2Y12R inhibitor and ATP

To block P2Y12R-mediated microglial actions, a P2Y12R inhibitor, PSB-0739 (#3983; Bio-Techne Corp.) dissolved in artificial cerebral spinal fluid (aCSF) (#3525, Bio-Techne Corp.; 40 mg/kg in 5 µl volume) was injected into the cisterna magna (CM) before surgery, while vehicle aCSF injection was used as a control. To mimic the focal increase of ATP, different concentrations of ATP were injected (0.3, 1.0, and 3.0 µg; Adenosine 5-triphosphate disodium salt hydrate, #A1852-1VL, Sigma) dissolved in purified water (Ambion™ Nuclease-Free water, #AM9937, Invitrogen), while vehicle purified water injection was used as a control. CM injections were done under 1–1.5% isoflurane anesthesia.

## Functional outcome test

All functional tests were performed blinded

### Neuroscore
The Neuroscore was performed before surgery, 4, 24, 48 h and 3 d after TIA; this test was used to evaluate the general status and focal neurologic dysfunction after TIA and was performed as described before (Orsini et al, 2012). The score ranges from 0 (no deficits) to 56 (representing the poorest performance in all items) and is calculated as the sum of the general and focal deficits. The Neuroscore results were expressed as a composite neurological score, which included the following general deficits (scores): fur (0–2), ears (0–2), eyes (0–4), posture (0–4), spontaneous activity (0–4), epileptic behavior (0–12); and the following focal deficits: body asymmetry (0–4), gait (0–4), climbing on a surface inclined at 45° (0–4), circling behavior (0–4), front-limb symmetry (0–4), circling behavior (0–4), and whisker response to light touch (0–4).

### Barnes maze
A modified version described before (Attar et al, 2013) was used to perform the Barnes maze test. The elevated 20-hole apparatus

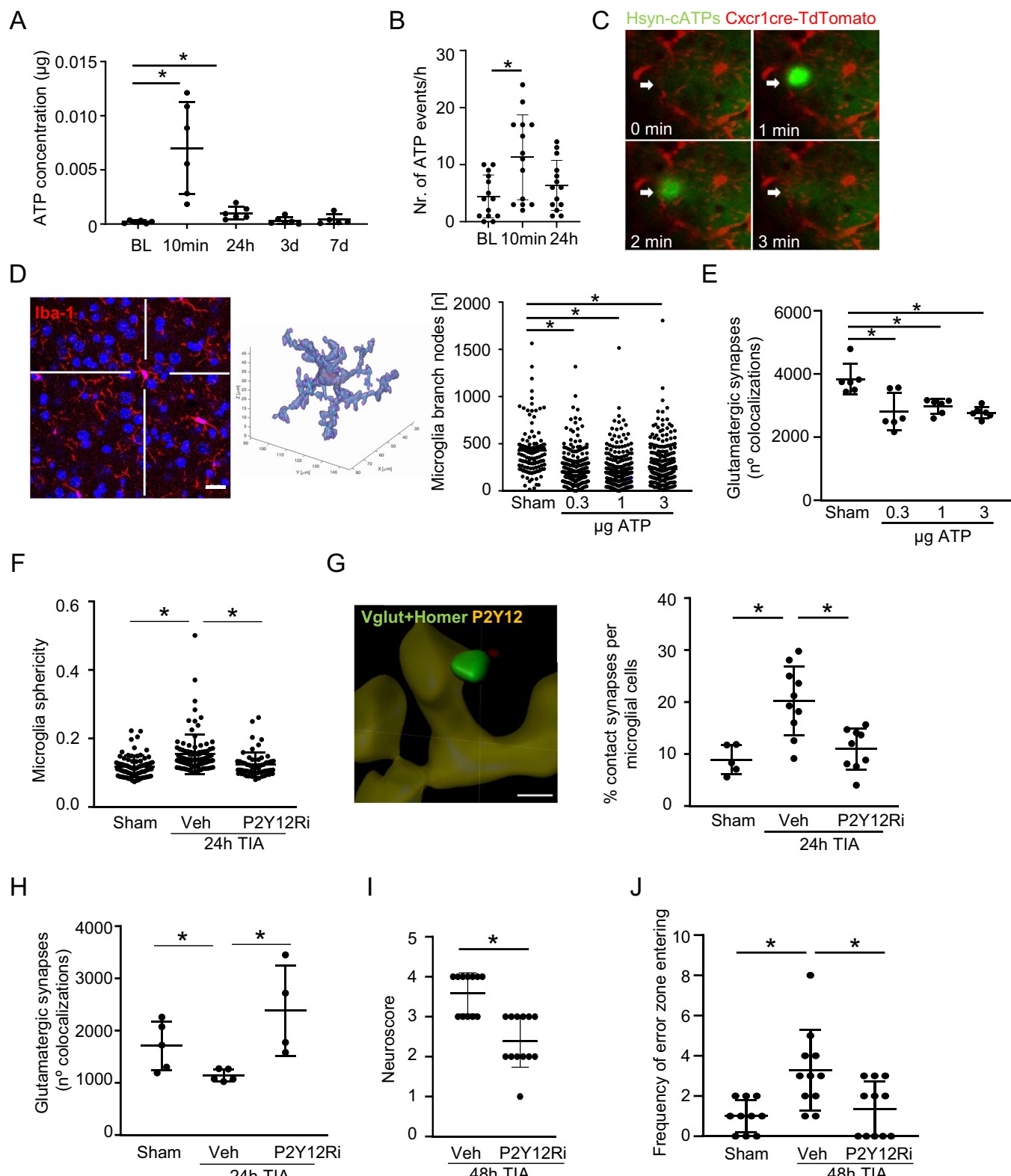

(diameter: 100 cm, hole diameter: 10 cm) had a target box that was placed under the maze. The protocol includes three phases of interaction of mice with the maze: (1) habituation, (2) 3-day training, TIA surgery and 48 h later (3) probe. Before each day of training or probe, mice were placed 30 min before the procedure in the testing room for acclimatization. On day 1, mice were habituated to the maze. Therefore, mice were placed in the center of the maze in a 2-l glass beaker. After 1 min of acclimatization,

**Figure 3. P2Y12R inhibition confirms ATP-mediated microglia activation after TIA.**

(A) Luciferace assay to detect ATP concentration from peri-lesional perfusate (see Fig. 2A) at before (BL) and at indicated time points after TIA in males ($n = 6$ per group; 10 min: $p$ value = 0.0002, 24 h: $p$ value = 0.0231). (B) Quantification of extracellular ATP (hsyn-cATPs) using in vivo two-photon imaging before (BL) and at indicated time points after TIA ($n = 18$ per group; 10 min: $p$ value <0.01) and (C) microglial processes (Cx3Cr1cre-TdTomato) recruitment to focal increase in extracellular ATP within 3 min after TIA in males. (D) Representative image of Iba1+ microglia cell and 3D reconstructed microglia for branches nodes analysis 24 h after cisterna magna injection of different ATP concentrations in males ($n = 6$ per group; 0.3, 1, and 3 μg ATP: $p$ value <0.0001, scale bar = 20 μm. (E) Quantification of colocalized presynaptic and postsynaptic particles 24 h after cisterna magna injection of different ATP concentrations in males ($n = 6$ per group; 0.3, 1, and 3 μg ATP: $p$ value <0.0001). (F) Microglia sphericity ($n = 5$ per group; Veh: $p$ value <0.0001 and P2Y12Ri: $p$ value <0.0001), (G) synapse (Vglut+Homer+) contacts with microglial cells (P2Y12+) ($n = 10$ per group; Veh: $p$ value <0.0033 and P2Y12Ri: $p$ value <0.0035) (scale bar = 2 μm) and (H) glutamatergic synapse counts ($n = 5$ per group) in sham and 24 h after TIA in vehicle (Veh) or P2Y12Ri (-inhibitor) treated animals (in males' mice); Veh: $p$ value = 0.0140 and P2Y12Ri: $p$ value = 0.0155. (I) Neuroscore at 48 h after TIA in vehicle (Veh) or P2Y12Ri (-inhibitor) treated animals ($n = 10$ per group; $p$ value = 0.0002) and (J) Barnes Maze test ($n = 10$ per group) in Sham and 48 h after TIA in vehicle (Veh) or P2Y12Ri (-inhibitor) treated animals (in males' mice); Veh: $p$ value = 0.0083 and P2Y12Ri: $p$ value = 0.0469. Statistical tests: (A–J) two-way ANOVA, corrected for multiple comparisons using two-stage step-up method of Benjamin Kriegel. (I) Mann–Whitney $U$-test. Error bars indicate ±SD. *$p$ < 0.05. Source data are available online for this figure.

mice were guided slowly by moving the glass beaker toward the target hole. On day 2, mice were placed in the center and given the possibility to freely explore and find the target hole. If mice did not reach the target hole within 2 min, the glass beaker was used to slowly guide them to the target. This was done in two consecutive trials. On days 3 and 4, mice were placed under the beaker and 10 s after placement, the beaker was removed, and mice were allowed to explore freely and find the target hole. Again, two consecutive trials were performed with every mouse. This procedure was repeated for the actual probe 3 d later. TIA surgery was performed on day 5 (after training). Acquisition and zone-dependent analysis were performed with Ethovision XT (Noldus).

## In vivo wide-field calcium imaging

To visualize calcium dynamics of cortical (L2/3) excitatory neurons, an optogenetic calcium-reporter mouse strain, C57BL/6J-Tg(Thy1- GCaMP6s)GP4.12Dkim/J (Dana et al, 2014) was used. First, an imaging window was created over the dorsal cranium as follows. The animals were anesthetized with inhalation anesthesia of isoflurane (5% for induction and 2% during operation) in 70% nitrous oxide and 30% oxygen and fixed in a prone position in a stereotaxic frame (Stoelting). Then the skin covering the skull and the underlying connective tissue were removed, and a layer of transparent dental cement (Parkell C&B Superbond) was distributed on the window area and covered with a manually fitted coverslip (Thermo Fisher). Afterward, the mice were allowed to recover from the surgery for more than 48 h before the first image acquisition.

For image acquisition, mice were injected with 0.05 mg/kg body weight of medetomidine intraperitoneally 5 min before inducing inhalation anesthesia with a mixture of 5% isoflurane in 70% nitrous oxide and 30% oxygen. After 70 s, the mice were secured in a stereotactic frame. Subsequently, the isoflurane concentration was reduced to 1.5% for 140 s, and then further decreased to 0.75% for 2 min to achieve steady state prior to data acquisition. In vivo wide-field calcium imaging was conducted using a custom-built imaging system, described by Cramer et al (2019). This system allowed mesoscale imaging of dorsal cortical brain areas across both hemispheres through the intact skull, encompassing a field of view of 10 × 10 mm. This corresponds to an image matrix resolution of 512 × 512 pixels. Image acquisition was performed over a 4-min period at a frequency of 50 Hz, capturing both fluorescent and background (no light) channels, in a room devoid of ambient light. For each subject, a total of six imaging sessions were conducted: baseline, during filament insertion, 4 h, 24 h, 3 d, and 7 d after filament insertion. Following the imaging session, anesthesia was reversed by administering Atipamezole at a dosage of 0.1 mg/kg body weight intraperitoneally. Throughout the anesthetization procedures, the body temperature of the animals was regulated using a feedback-controlled heating system. After the completion of the surgical interventions, the animals were placed in a heating chamber until full recovery from the anesthesia was observed.

Image acquisition, image processing and the calculation of functional connectivity, were conducted as previously described (Cserep et al, 2020). Briefly, subsequent to motion correction and image alignment, the background was removed from each fluorescent image. The delta F/F transformation was then applied by centering and normalizing the fluorescent images to the record-wide average for each pixel. Functional connectivity was assessed by calculating Pearson correlation coefficients for the signal time courses, followed by Fisher z-transformation, across 16 predefined regions of interest (ROIs) that represent distinct anatomical and functional brain areas. Average connectivity scores were grouped, and differences between groups were depicted for all ROI pairs using a heat map. Overall functional connectivity alterations due to TIA were evaluated by computing the global connectivity (GC) for each pixel. To assess the effects of TIA, GC scores were averaged pixel-wise within group (TIA and Sham group). The intergroup differences were then visualized using a topographical map representing all brain pixels. Network connectivity was analyzed in Matlab version 2016b, visualized in R version 4.3.1 and evaluated by repeated measures of ANOVA followed by Turkey's post hoc test. Differences with a $p$ value <0.05 were considered to be statistically significant.

## In vivo two-photon imaging

For [ATP]$_e$ measurements, animals were injected with a green ATP sensor (#YL006006-AV9, AAV9-hsyn-cATP1.0(chick), WZ Biosciences, Inc., Columbia, MD, USA) 1 week before cranial window surgery. Cranial window (3 mm, circular) was placed over the primary somatosensory cortex of CX3CR1$^{GFP/+}$or CX3CR1$^{tdTomato}$ mice, and 2–3 weeks after surgery, [ATP]$_e$ events were imaged in awake animals using the resonant-scanning light path at 960 or 920 nm with 16× water-immersion objective (Nikon CFI75 LWD

16× W, NA 0.8) at 32.7521 Hz. The acquisitions were performed before and 1 h long after TIA, the recordings typically started 5 min after TIA (5 min fMCAo). Repetitively, 2–3 min long videos (with 2 min pauses) were recorded from the same FOVs. ATP events were manually quantified for every animal. For microglial process motility measurements, the galvo light path was used at 920 nm or 1040 nm. Z-stacks (consisting of six individual planes) were recorded every minute during the 30 min baseline and 1 h after TIA sessions from anesthetized (phentanyl) mice with the 16x objective. Thirty individual microglial processes per FOV were manually tracked in 2D with FIJI, and maximum, average and median process speed were calculated according to the FIJI coordinates in Excel. Imaging was performed on a Femto2D-DualScanhead microscope (Femtonics) coupled with a Chameleon Discovery laser (Coherent). Data acquisition and analysis were performed with MESc (v.3.5.6.9395SLE) and MES (v.5.3560) software (Femtonics, Budapest, Hungary).

## [18 F]FDG positron-emission tomography (PET)

The mice were scanned before, 24 h, 3 and 7 d after the TIA induction using a 3T Mediso nanoScan PET/MR scanner (Mediso Ltd, Hungary) with a single-mouse imaging chamber. The mice received an intravenous injection of $18.0 \pm 2.1$ MBq[18 F]FDG through the tail vein. For the dynamic PET imaging (3TIA and 3sham mice), acquisition was performed from 0 to 90 min after tracer injection (analysis cohort). For the static PET imaging (3TIA and 3sham mice), the list-mode data were acquired at 60–90 min after tracer injection (validation cohort). A 15-min anatomical T1 MR scan was performed at 30 min after[18 F]FDG injection for the validation cohort (static imaging) and after 90 min for the analysis cohort (head receive coil, matrix size $96 \times 96 \times 30$, voxel size $0.21 \times 0.24 \times 0.65$ mm$^3$, repetition time 677 ms, echo time 28.56 ms, flip angle 90°). The PET field of view (FOV) included the whole mouse, while the MRI FOV covered the mouse head only. The T1 image was then used to create a body-air material map for the attenuation correction of the PET data. We reconstructed the PET list-mode data within a 400–600 keV energy window using a 3Diterative algorithm (Tera-Tomo 3D, Mediso Ltd, Hungary) with the following parameters: matrix size $55 \times 62 \times 187$ mm$^3$, voxel size $0.3 \times 0.3 \times 0.3$ mm$^3$, eight iterations, six subsets. When acquired dynamically (0–90 min p.i. acquisitions), the list-mode data were binned into 25 frames ($6 \times 10$, $2 \times 30$ s, $3 \times 1$, $5 \times 2$, $5 \times 5$, $5 \times 10$min). Decay and random correction were applied.

Native space attenuation- and motion-corrected PET images were averaged and affine registered to an in-house mouse FDG-PET template and intensity normalized to the mean tracer uptake in the cerebellum. Mean intensity-normalized SUVR values were then extracted for Regions of interest included in the Mirrione atlas (Mirrione et al, 2007).

## Assessment of apoptotic cells (TUNEL)

Mice were deeply anesthetized 24 h after TIA induction with ketamine (120 mg/kg) and xylazine (16 mg/kg) and transcardially perfused with 10 ml saline. Brains were removed and frozen immediately on powdered dry ice and stored at −20 °C. About 20-μm-thick coronal sections were obtained at the level of somato-sensory hindlimb (Bregma: 0.5 mm posterior). Cell death was measured with the terminal deoxynucleotidyl transferase-mediated dUTP nick-end labeling (TUNEL) method (TUNEL apoptosis detection kit; Millipore, #S7110). Finally, sections were stained with DAPI and mounted with fluoromount medium (Sigma). Two images, per animal, were taken at 40x magnification on a confocal microscope (Zeiss 880) in the striatum region, and TUNEL-positive cells were counted using FIJI software (Analyze particles).

## Tissue hypoxia staining (Hypoxyprobe)

Hypoxic tissue was detected using Hypoxyprobe-1 Green Kit (HPI, #HP6-XXX). Briefly, mice were administered with intraperitoneal injection of 60 mg/Kg Pimonidazole Hydrochloride and after 25 min mice were deeply anaesthetized with ketamine (120 mg/kg) and xylazine (16 mg/kg) and transcardially perfused with 10 ml normal saline and 10 ml 4% PFA (pH 7.4), then brains were removed and post-fixed in 4% PFA for 18 h at 4 °C. About 100-μm-thick coronal sections were obtained at the level of the somatosensory hindlimb (Bregma: 0.5 mm posterior). Brain sections were incubated with FITC-antipimonidazole Mab1 (1:100, overnight). Finally, sections were stained with DAPI and mounted with fluoromount medium (Sigma). Samples were analyzed on a confocal microscope (Zeiss 880).

## Blood–brain barrier integrity analysis

Alterations in blood–brain barrier integrity was evaluated by the Evans blue (EB) assay. Briefly, mice were administered with intraperitoneal injection of 200 µl EB solution (1% in PBS, 2.5 ml/kg; Sigma-Aldrich) and after 2 h, mice were deeply anaesthetized with ketamine (120 mg/kg) and xylazine (16 mg/kg) and transcardially perfused with 10 ml normal saline. Brains were quickly removed, weighed, minced into small pieces and incubated in 500 µl formamide (Roth) for 24 h at 55 °C. Samples were then centrifuged for 20 min at 10,000×$g$, and supernatants from each sample were transferred to a clear flat-bottom 96-well plate. The optical density of the samples was measured in duplicate on a spectrophotometer plate reader at 620 nm (Bio-Rad), and the amount of EB in the samples was quantified using the standard curve method.

## Immunohistology

Mice were transcardially perfused at the indicated time points with 10 ml saline and 10 ml 4% PFA (pH 7.4), then post-fixed in 4% PFA for 18 h at 4 °C.100-μm-thick coronal sections were obtained at the level of somatosensory hindlimb (Bregma: 0.5 mm posterior) for immune histochemical analysis. Free-floating sections were frozen in cryoprotectant and stored at −80 °C. Then sections were washed in PBS overnight at 4 °C. After washing with PBS, sections were incubated in blocking buffer containing 0.1% Triton, 0.05% Tween 20, 1% bovine serum albumin, 0.1% cold fish skin gelatine, and 2% goat or donkey serum in PBS at RT for 1 h. For the glutamatergic synapse staining, sections were incubated for 3 d at 4 °C with Guinea pig anti-VGlut1 antibody (1:1000, Millipore #AB5905), Chicken anti-Homer (1:2000, Synaptic Systems #160006) and labeled overnight at 4 °C with the secondary antibody AF488 goat anti-Guinea pig (1:500, Invitrogen), AF647 goat anti-Chicken (1:500, Invitrogen). For neuron-microglia interaction staining,

sections were incubated overnight at 4 °C with rabbit anti-P2Y12 antibody (1:200, AnaspecInc#AS-55043A), mouse anti-KV2.1 (1:200, NeuroMab#75-014) and labeling for 1 h at room temperature with the secondary antibody AF488 goat anti-rabbit (1:200, Invitrogen A11034), AF647 goat anti-mouse (1:200, Invitrogen #A21235). For microglia morphology analysis staining, sections were incubated overnight at 4 °C with rabbit anti-Iba1 (1:100, Wako #019-19741) and labeling for 2 h at room temperature with the secondary antibody AF594 goat anti-rabbit (1:200, Invitrogen). For neuronal staining, sections were incubated overnight at 4 °C with mouse anti-NeuN antibody (1:100, MerkMillipore MAB377) and labeling for 1 h at room temperature with the secondary antibody AF488 goat anti-mouse (1:200, Invitrogen A11001). Finally, sections were stained with DAPI and mounted with fluoromount medium (Sigma). Two images, per animal, were taken at 40x magnification on a confocal microscope (Zeiss 880) in the cortex region, just above the hippocampus.

For the synapse analysis, confocal Z-stack images were processed, and colocalizations of VGlut and Homer were analyzed using ImageJ (synapse counter plugin).

For the microglia-synapse analysis, confocal Z-stack images were processed, and colocalizations of synapses (VGlut+ Homer+) in contact with microglial cells (P2Y12+) were analyzed using Imaris software. A 3D reconstruction for synapses (VGlut+ Homer +) and a 3D reconstruction for microglial cells (P2Y12+) were generated. Synapses in microglial cells were counted as engulfed synapses, and synapses partially in microglial cells were counted as contacts.

For the microglia–Neuron analysis, confocal Z-stack images were obtained and analyzed using ImageJ software. To measure the distribution of Kv2.1 labeling relative to microglial processes, confocal stacks were exported into single-channel TIFF series. Cell volume of eight random pyramidal neurons (KV2.1+) per image were calculated frame by frame using FIJI. Then the microglial contacts (P2y12+) to each neuron was calculated frame by frame, and the different surface contacts were calculated per each pyramidal neuron using FIJI.

For the microglia morphology analysis, confocal Z-stack images were processed, and microglial morphology features were extracted using custom-written scripts in MATLAB (R216b, The Math-Works, Natick, Massachusetts, USA), with dependencies on the Image Processing Toolbox as well as Statistics and Machine Learning Toolbox. Statistical analysis and data visualizations were performed in RStudio4 using R version 3.2.2 5 and the packages ROCR 6, plyr7, beeswarm8, and corrplot9. The detailed protocol and properties of the MATLAB script have been previously described (Heindl et al, 2018). A Kruskal-Wallis test with post hoc Bonferroni correction was applied for multilevel comparisons between groups.

## Golgi-Cox staining and dendritic spine analysis

Following saline perfusion, mice were perfused with aldehyde fixative solution (Bioenno, #003780). Brains were then carefully removed and placed in fixative solution at 4 °C overnight. Brains were then sliced at 100-μm-thick vibratome sections and immersed in impregnation solution (Bioenno, sliceGolgiKit, #003760) for 5 d. Further staining was performed as described by the manufacturer (Bioenno). In total, five dendrites from five

neurons each were imaged (100× brightfield). Dendrites from the images were then reconstructed using Imaris ×64 (version 8.4.0, Bitplane).

## Cerebral open flow microperfusion (cOFM)

Brain extracellular tissue fluid was collected before and at different time points after TIA surgery using cerebral open flow microperfusion (cOFM) as previously described (Birngruber et al, 2013). Briefly, mice were anesthetized, and their head was fixed in a stereotactic frame. A 15 mm skin incision was made to expose the skull. The cOFM probe (cOFM P1-1, Joanneum research) was inserted slowly to a depth of 1 mm into the somatosensory cortex (Bregma: 1.5 lateral, 0.5 mm posterior) via a 1 mm hole drilled into the skull. The probe was fixed to the skull using a biocompatible dental cement (Quick Base S398, L-Powder clear S399, Universal Catalyst S371, Parkell C&B metabond, USA). The composition of the standard cOFM perfusate was designed to match brain extracellular fluid to avoid chemical stress. The specific composition of artificial cerebrospinal fluid (aCSF) was prepared as described in the Alzet webpage (https://www.alzet.com/guide-to-use/preparation-of-artificial-csf/): NaCl 148.18 mM; MgCl$_2$ 0.8 mM (purity ≥98%); CaCl2 1.4 mM (purity ≥93%); KCl 3mM; NaH$_2$PO$_4$ 0.19 mM; Na$_2$HPO$_4$ 1.2 mM; glucose 3.7 mM; urea (Carbamide) 6.7 mM. All reagents were dissolved in sterile water (Aqua bidest; Fresenius Kabi, Graz, Austria). The perfusate was filtered through a 0.22-ml sterile filter (Thermo Fisher Scientific, Schwerte, Germany). All steps were performed under sterile conditions. Three weeks after cOFM insertion, the healing dummy was replaced by the inflow and outflow tubing before sampling; perfusate is then injected into the brain tissue and withdrawn at the same flow rate. The inflow and outflow tubing are connected to a micropump at a flow rate of 0.8 μl/min.

## Proteomics sample preparation

Brain extracellular tissue fluid was collected for 60 min using cOFM and immediately frozen on dry ice, before and at 24 h, 3 d, and 7 d after TIA.40 μl of mouse brain fluid dialysates were prepared according to the single-pot solid-phase-enhanced sample preparation (SP3) protocol (Hughes et al, 2019) with some adaptations. In brief, samples were mixed 1:1 with lysis buffer containing 10% SDS in PBS, pH 8.5, 10 mM Tris (2-carboxyethyl) phosphine, and 30 mM 2-chloroacetamide. Proteins were denatured, reduced, and alkylated by incubating samples at 70 °C, 800 rpm on a shaker for 10 min. We added 2 μl of prepared magnetic bead mixtures (Sera-Mag Magnetic carboxylate modified particles, Cytiva) to each sample and mixed them 1:1 with 100% acetonitrile (ACN). Samples were incubated for 8 min at RT and 800 rpm on a shaker. Following, samples were transferred on a magnet, and magnetic beads with bound proteins were washed two times with 200 μl 70% ethanol and once with 100% ACN while keeping the samples the whole time on the magnet. For enzymatic digestion, the magnetic beads were resuspended in 10 μl digestion buffer containing 0.1 μg trypsin/Lys-C mixture (Promega) in 50 mM ammonium bicarbonate buffer, pH 8.0 and digestion at 37 °C, 400 rpm overnight. The next day, samples were mixed with 380 μl 100% ACN and incubated for 8 min at room temperature. After transfer on a magnet, the beads were washed with 200 μl 100% ACN and air-

dried. Peptides were eluted from magnetic beads by resuspending the same in 9 μl of 5% ACN in LC-MS-grade water. We determined peptide concentrations and used 250 ng peptides per sample for each LC-MS run.

## LC-MS/MS analysis

Samples were measured by liquid chromatography-tandem mass spectrometry using an Easy nLC 1200 chromatographic system (Thermo Fisher Scientific) and the Exploris 480 mass spectrometer (Thermo Fisher Scientific). Peptides were separated by 90 min chromatographic gradients using a binary buffer system with buffer A (0.1% formic acid in LC-MS-grade water) and buffer B (80% ACN, 0.1% formic acid in LC-MS-grade water). We used an in-house packed analytical column with a length of 50 cm and filled with 1.9 μm ReproSil-Pur 120 C18-AQ material (Dr. Maisch). To separate peptides, the amount of buffer B was linearly increased from 4 to 25% over 70 min with a constant flow rate of 300 nl/min. Following, buffer B was increased to 55% over 8 min and a sharp increase to 95% buffer B over 2 min. The analytical column was washed at 95% for 10 min.

Eluting peptides were ionized by nano-electrospray ionization at a constant spray voltage of 2.5 kV. We measured the samples in data-independent acquisition (DIA) mode. In brief, full MS were recorded at a resolution of 60,000 with an AGC target of 300% and a maximum injection time of 55 ms. The scan range was set to 340–1080 m/z. DIA MS/MS scans were recorded at a resolution of 15,000, an AGC target of 1000%, and a maximum injection time of 22 ms. The DIA window m/z range was set from 400–1000 m/z separated into 50 isolation windows with a size of 12 m/z per window. We used a staggered window approach with isolation windows shifted by 6 m/z every second scan cycle. Fragment ions were generated with an HCD collision energy of 27%. We measured pooled samples for the generation of a gas phase fractionation (GPF) spectral library. Similar LC-MS methods were applied, only the DIA scan ranges were limited to 100 m/z, covering the overall m/z range from 400–1000 m/z in six consecutive runs and staggered DIA windows with a size of 4 m/z. The mass spectrometry proteomics data have been deposited to the ProteomeXchange Consortium (http://proteomecentral.proteomexchange.org) via the PRIDE partner repository (https://academic.oup.com/nar/article/50/D1/D543/6415112?login=false) with the dataset identifier PXDxxxxxx.

## Statistical analysis of proteomics data

Staggered windows were deconvoluted with the MSConvert tool of the ProteoWizard software suit (v. 3.0.21321, (Chambers et al, 2012)). Spectral library generation and peptide identification/quantification from LC-MS raw data was performed with the DIA-NN software suit (v.1.8, (Demichev et al, 2020)). We used the SWISS-PROT Mus musculus fasta database downloaded from UniProt (v. 2021-11-18) to make a spectral library using the six GPF measurements from pooled samples. Trypsin was set as the digestion enzyme with a maximum of one miss-cleavage, and cysteine carbamidomethylation was set as a fixed modification. The scan window radius was set to 10, mass accuracies were fixed to 2e-05 (MS2) and 7.5e-06 (MS1), respectively. Precursor peptides were filtered at an FDR <1%. Label-free normalization of protein groups was performed in R using the MaxLFQ algorithm (Cox et al, 2014) and proteotypic peptides only.

Statistical analysis of the data were performed with the Perseus software suite (v. 1.6.15 (Tyanova et al, 2016)). In brief, we log2-transformed protein LFQ intensities and filtered proteins for data completeness in at least one time point. Missing values were replaced sample-wise by random drawing of numbers from 1.8 standard deviations downshifted, and 0.3 standard deviations broad normal distributions. Quantile normalization was performed with an R Perseus-plugin as well as a batch correction with ComBat R Perseus-plugin (batch = animal). Significantly regulated proteins were identified by ANOVA multiple sample testing (S0 = 0.1, permutation-based FDR = 0.05, 250 randomizations). ANOVA-significant proteins were further Z-score normalized and used for hierarchical clustering using Euclidean distances. We searched systematically for enriched processes in identified clusters using Fisher's exact testing (Benjamini–Hochberg FDR = 0.02).

## ATP luciferase assay

ATP concentration was determined using a luciferase assay as previously described (Sebastian-Serrano et al, 2018). Brain extra-cellular tissue fluid was collected for 15 min using cOFM and placed in a tube with 1 μl of 100 mM ARL 67156 (ARL 67156 trisodium salt hydrate, #A256-5MG, Sigma), a competitive inhibitor of ecto-ATPases and immediately frozen on dry ice to further determine the ATP concentration. The nucleotide concentration in the brain extracellular tissue fluid was measured using ENLITENR rLuciferase/Luciferin reagent (#FF2021, Promega) according to the manufacturer's protocol. Briefly, 2 μl of brain extracellular tissue fluid was transferred to wells of a 96-well plate placed on ice. The 96-well plate was set in a Glomax-multi Microplate Luminometer (Promega GmbH), and 100 mL of rLuciferase/Luciferin reagent was automatically injected into each well at room temperature.

## Cell isolation for RNA sequencing and loading Chromium Next GEM Chip G

Mice were perfused with ice-cold saline, and brains were gently removed. The hemispheres were split, and the white matter and meninges were removed with forceps. The tissue was dissociated using the papain dissociation system (#LK003150, Worthington Biochemical Corporation) according to the manufacturer's protocol and an incubation time of 15 min. The cell suspension was cleared from dead cells and cell debris using the dead cell removing kit (#130-090-101, Miltenyi). Propidium iodide-negative live single cells were flow cytometrically sorted using a BD Fusion cell sorter into RPMI medium containing 5% fetal bovine serum. The cell suspension was centrifuged and washed with PBS containing 0.04% bovine serum albumin (BSA). The cells were resuspended in PBS containing 0.04% BSA for a concentration of 1200 cells/μl, counted and loaded into the Chromium Next GEM Chip G (PN-1000127, 10x Genomics) for a target recovery of 10,000 cells according to the 10x Genomics Next GEM Single Cell 3' v3.1 (Dual Index) kit protocol (Step 1, CG000315, Rev E) for GEM preparation (PN-1000296, 10x Genomics) and run in a 10x Genomics Chromium Controller. After GEM preparation, the protocol of the kit was followed for post-GEM-RT Cleanup & cDNA amplification (Step 2)

and 3′ gene expression library construction (Step 3). Single-cell libraries were sequenced using an Illumina HiSeq 1500 sequencer with a sequencing depth of >20,000 reads per cell.

## Single-cell transcriptomic data analysis

Samples were demultiplexed, processed and aligned to the reference genome (mm10, GENCODE vM23/Ensembl 98), and unique molecular identifier (UMI) counts were summarized using CellRanger software (10x Genomics, v. 7.1.0). Filtered gene-barcode matrices were used for further analysis and processed using the Seurat package in R, version 4.3.0 (Hao et al, 2021). Cells that expressed fewer than 500 genes and contained more than 7% of mitochondrial genes were excluded, and genes expressed in less than three cells were removed from the data count matrices. After quality control, the data were log-normalized, and variable features were identified using the *vst* method. Scaling and regression against the number of UMIs and mitochondrial RNA content per cell was applied. Unbiased clustering was applied using k-nearest neighbor calculation, and dimensionality reduction was performed using PCA and UMAP. Differentially expressed genes among the clusters were identified using the FindAllMarkers function of the Seurat package. Figure 2E shows absolute gene counts of the preprocessed scRNAseq dataset per cell group for time points 24 h and 3 d after TIA induction compared to the control condition. Since the differential gene expression analysis was performed as pseudo-bulk using the function "FindAllMarkers()" in Seurat, which uses average expression over all cells.

Differentially regulated genes per cell type (Fig. EV4A) were calculated from log-normalized expression values from raw gene counts, which were then log-normalized and scaled (*z*-score). For the barplot, all genes with a *p* value <0.05 for differential expression were included. Gene set enrichment analysis (GSEA) was performed based on these significantly differentially regulated genes using ClusterProfiler (Version 4.7.2). All analysis scripts will be made available upon publication.

## Microglia transcriptomics (Nanostring)

Mice were perfused transcardially with ice-cold saline containing Heparin (2 U/mL). Brains were placed in HBSS (w/divalent cations $Ca^{2+}$ and $Mg^{2+}$) supplemented with actinomycin D (1:1000, 1 mg/mL, Sigma, #A1410), and microglia was isolated with the Papain-based Neural Tissue Dissociation Kit (P) (# 130-092-628, MiltenyiBiotec B.V. & Co. KG) according to the manufacturer's instructions. Cell suspension was enriched using a 30% isotonic Percoll gradient. $1 \times 10^3$–$1.5 \times 10^3$ live microglia-like cells from three mice per condition were sorted according to their surface marker CD45 + CD11b + 7-AAD negative (SH800S Cell Sorter, Sony Biotechnology). Microglia-like cells' total RNA was extracted using the Arcturus PicoPure RNA Isolation Kit (Applied Biosystems #KIT0204). About 65 ng of total RNA per sample was then hybridized with reporter and capture probes for Counter Gene Expression code sets (Mouse Neuroinflammation code set) according to the manufacturer's instructions (NanoString Technologies). Samples were injected into the NanoString cartridge and a measurement run was performed according to the nCounter SPRINT protocol. Background (negative control) was quantified by code set intrinsic molecular color-coded barcodes lacking the RNA

linkage. As a positive control code set, intrinsic control RNAs were used at increasing concentrations. Genes below the maximal values of the negative controls were excluded from the analysis. Data was analyzed by ROSALIND® (https://rosalind.bio/), with a HyperScale architecture developed by ROSALIND, Inc. (San Diego, CA). Read Distribution percentages, violin plots, identity heatmaps, and sample MDS plots were generated as part of the QC step. Normalization, fold changes and *p* values were calculated using criteria provided by Nanostring. ROSALIND® follows the nCounter® Advanced Analysis protocol of dividing counts within a lane by the geometric mean of the normalizer probes from the same lane. Housekeeping probes to be used for normalization are selected based on the geNorm algorithm as implemented in the NormqPCR R library (Perkins et al, 2012). Abundance of various cell populations is calculated on ROSALIND using the Nanostring Cell Type Profiling Module. ROSALIND performs a filtering of Cell Type Profiling results to include results that have scores with a *p* value greater than or equal to 0.05. Fold changes and *p* values are calculated using the fast method as described in the nCounter® Advanced Analysis 2.0 User Manual. *P* value adjustment is performed using the Benjamini–Hochberg method of estimating false discovery rates (FDR). Significantly regulated genes were identified by ANOVA multiple sample testing (S0 = 0.1, permutation-based FDR = 0.05). ANOVA-significant genes were further *Z*-score normalized and used for hierarchical clustering using Euclidean distances, using the Next Generation Clustered Heat Map (NG-CHM) software (GUI Builder 2.22.0 w/ NG-CHM 2.24.0-build-20)(Ryan et al, 2019). We searched systematically for enriched processes in identified clusters using Fisher's exact testing (Benjamini–Hochberg FDR = 0.05). Further, we performed enrichment analyses (Benjamini–Hochberg FDR = 0.05) on fold changes calculated for binary comparisons(Kolberg et al, 2023).

## Platelet assays

For platelet isolation, anesthetized mice underwent blood collection via insertion of a glass capillary into the retro-orbital venous plexus. Blood was collected into tubes containing 1/7 volume of acid-citrate-dextrose (ACD; 39 mM citric acid, 75 mM sodium citrate, and 135 mM dextrose) and immediately diluted 1:1 with modified Tyrode's buffer (137 mM NaCl, 2.8 mM KCl, 12 mM $NaHCO_3$, 5.5 mM sucrose, 10 mM HEPES, and pH 6.5). Samples were centrifuged at 70×*g* for 15 min to obtain platelet-rich plasma (PRP). For preparation of washed platelets, PRP was diluted 1:2 in modified Tyrode's buffer supplemented with prostacyclin ($PGI_2$, 0.1 mg/ml, Abcam) and casein (0.01%, Sigma), followed by centrifugation at 1000×*g* for 5–10 min. The platelet pellet was resuspended in Tyrode's buffer, and platelet counts were determined using a Sysmex XN-V Series XN-1000V cell counter.

For platelet aggregation, optical aggregometry was performed as previously described by our collaborators (Nicolai et al, 2020). Briefly, washed murine platelets were adjusted to a final concentration of $2 \times 10^5$/µl in modified Tyrode's buffer. Platelet activation was induced by the addition of adenosine diphosphate (ADP; 20 µM, Sigma, #01905) targeting the P2Y12 receptor, together with calcium chloride (2 mM), under continuous stirring at 1000 rpm at 37 °C. Aggregation was recorded using a two-channel aggregometer (ChronoLog490-2D, Havertown, USA), and

## The paper explained

### Problem

Transient ischemic attacks (TIAs) are traditionally considered benign because they do not cause structural brain damage. Nevertheless, many patients develop persistent cognitive impairment and an increased risk of dementia. The biological mechanisms linking brief cerebral ischemia to lasting neurological dysfunction remain largely unknown.

### Results

By establishing a novel mouse model that fulfills the tissue-based definition of TIA, this study shows that brief ischemia induces long-lasting cortical network disruption and synapse loss without neuronal death. Single-cell and proteomic analyses identified microglia as the primary responders after TIA. A rapid surge in extracellular ATP triggered excessive microglial process motility and increased microglia–neuron and synaptic pruning by reactive microglia via the purinergic receptor P2Y12. Pharmacological inhibition of P2Y12 normalized the microglial phenotype, preserved synaptic integrity, and prevented post-TIA cognitive deficits without affecting systemic platelet function.

### Impact

These findings redefine TIA as a condition with active neuroimmune pathology rather than a purely transient vascular event. Targeting microglial purinergic signaling emerges as a promising strategy to prevent secondary cognitive decline after TIA, opening new therapeutic avenues for a highly prevalent but underexplored cerebrovascular disorder.

the maximal aggregation response was analyzed 6 min after agonist addition using Aggrolink software (ChronoLog, USA).

Platelet activation in suspension was assessed as previously described by our collaborators (Nicolai et al, 2020). Washed murine platelets ($2 \times 10^5/\mu l$) were diluted 1:10 in modified Tyrode's buffer, activated by the addition of calcium (1 mM) and ADP (20 μM), and incubated for 30 min at room temperature with fluorescently labeled antibodies targeting the activation markers P-selectin (JonA-PE, #D200, Emfret Analytics) and activated GPIIbIIIa (CD62P-PE/Cy7, #148310, BioLegend). Following activation, platelets were fixed with 1% paraformaldehyde (PFA, Sigma) for 10 min in the dark. Samples were analyzed using a BD LSRFortessa flow cytometer, and subpopulation gating as well as mean fluorescence intensity (MFI) analyses were performed using FlowJo software (version 10).

## Statistical analysis

Data were analyzed using GraphPad Prism version 6.0. Summary data are expressed as the mean ± standard deviation (SD) for all datasets. All datasets were tested for normality using the Shapiro–Wilk normality test. The groups containing normally distributed data were tested using a two-way Student's $t$-test (for two groups) or ANOVA (for >2 groups). The remaining data were analyzed using the Mann–Whitney $U$-test. Neuroscore data were treated as non-parametric data. Two-way ANOVA, corrected for multiple comparisons using two-stage step-up method of Benjamin Kriegel and Yekutielior Mann–Whitney $U$-test have been used in the analysis. Differences with a $p$ value $<0.05$ were considered to be statistically significant.

## Data availability

The datasets and computer code produced in this study are available in the following databases: scRNAseq data: Gene Expression Omnibus GSE272954. Nanostring data: Gene Expression Omnibus GSE312513. Proteomics data: ProteomeXchange PXD070754. Immunofluorescent images: Bioimage archive S-BIAD2496 (Fig. 1E), S-BIAD2497 (Fig. 1F), S-BIAD2498 (Fig. 2H), S-BIAD2499 (Fig. 2I), S-BIAD2500 (Fig. 3D), S-BIAD2501 (Fig. 3E), S-BIAD2502 (Fig. 3F), S-BIAD2503 (Fig. 3G), S-BIAD2504 (Fig. 3H).

The source data of this paper are collected in the following database record: biostudies:S-SCDT-10_1038-S44321-026-00397-6.

## Peer review information

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

## Acknowledgements

This work was funded by the Deutsche Forschungsgemeinschaft (DFG, German Research Foundation) under Germany's Excellence Strategy within the framework of the Munich Cluster for Systems Neurology (EXC 2145 SyNergy–ID 390857198), through CRC1744 (ID 548585053) and under the grants LI-2534/6-1, LI-2534/7-1, and LL-112/1-1. The authors would like to thank Kerstin Thuß-Silczak, Sarah Slamova, and Christina Bauer for their excellent technical assistance.

## Author contributions

**Gemma Llovera**: Conceptualization; Formal analysis; Investigation; Visualization; Writing—original draft; Writing—review and editing. **Steffanie Heindl**: Formal analysis; Investigation; Visualization. **Daniel P Varga**: Resources; Formal analysis; Investigation; Visualization. **Nikolett Lenart**: Resources; Formal analysis; Investigation. **Sebastian Kallabis**: Formal analysis; Investigation. **Vanessa Göb**: Resources. **David Stegner**: Resources. **Raphael Escaig**: Resources; Formal analysis; Investigation. **Leo Nicolai**: Resources. **Nicolai Franzmeier**: Formal analysis. **Felix Meissner**: Resources. **Adam Denes**: Resources; Supervision; Visualization. **Arthur Liesz**: Conceptualization; Resources; Supervision; Funding acquisition; Visualization; Writing—original draft; Writing—review and editing.

Source data underlying figure panels in this paper may have individual authorship assigned. Where available, figure panel/source data authorship is listed in the following database record: biostudies:S-SCDT-10_1038-S44321-026-00397-6.

## Funding

## Disclosure and competing interests statement

The authors declare no competing interests.

# Expanded View Figures

**Figure EV1.   Characterization of the TIA model.**

General and focal Neuroscore at different time points after TIA in (**A**) males ($n = 15$ per group; General Neuroscore: 4 h: *p* value $= 0.0004$, 24 and 48 h:*p* value $<0.0001$; Focal Neuroscore: 4, 24, and 48 h:*p* value $<0.0001$, 3 d:*p* value $= 0.0012$) and (**B**) females ($n = 5$ per group; General Neuroscore: 4, 24, 48 h and 3 d:*p* value $<0.0001$; Focal Neuroscore: 4, 24 h: *p* value $<0.0001$, 48 h:*p* value $= 0.0002$, 3 d:*p* value $= 0.045$). (**C**) Representative image for brain hypoxia (hypoxyprobe in red) and cell death (TUNEL+ in green) 24 h after TIA in males in cortex (1), hippocampus (2) and striatum (3), scale bar $= 20$ μm. (**D**) Representative TUNEL+ staining and quantification 24 h after different MCA occlusion times in males ($n = 5$ per group), scale bar $= 20$ μm. (**E**) Blood–brain barrier integrity was measured by extravasated Evans Blue per gram of the brain tissue 24 h after different ischemia times in males. Black $=$ ipsilateral hemisphere, red $=$ contralateral hemisphere ($n = 5$ per group; 15 and 30 min ischemia: *p* value $= 0.0079$). (**F**) Representative image and analysis of neural cells (NeuN+) in different brain regions in males (DG dentate gyrus), scale bar $= 20$ μm. (**G**) Representative image and analysis of glucose metabolism using in vivo positron-emission tomography 24 h after TIA or Sham males ($n = 6$ per group). 1-Right striatum, 2-Left striatum, 3-Cortex, 4-Right hippocampus, 5-Left hippocampus, 6-Thalamus, 7-Carebelum, 8-Basal forebrain/septum, 9-Hypothalamus, 10-Right amygdala, 11-Left amygdala, 12-Brainstem, 13- Central gray, 14-Superior colliculi, 15-Olfactory bulb, 16-Midbrain right, 17-Midbrain left, 18- Inferior colliculi left, 19-Inferior colliculi right. (**H**) Representative scheme of the microperfusion system and protein intensity over time before (BL) and after TIA in males ($n = 5$ per group). Statistical tests: (**A, B, E, F**) two-way ANOVA, corrected for multiple comparisons using two-stage step-up method of Benjamin Kriegel. (**H**) Two-way ANOVA, Fisher's exact testing. Error bars indicate ±SD. *$*p < 0.05$. Source data are available online for this figure.

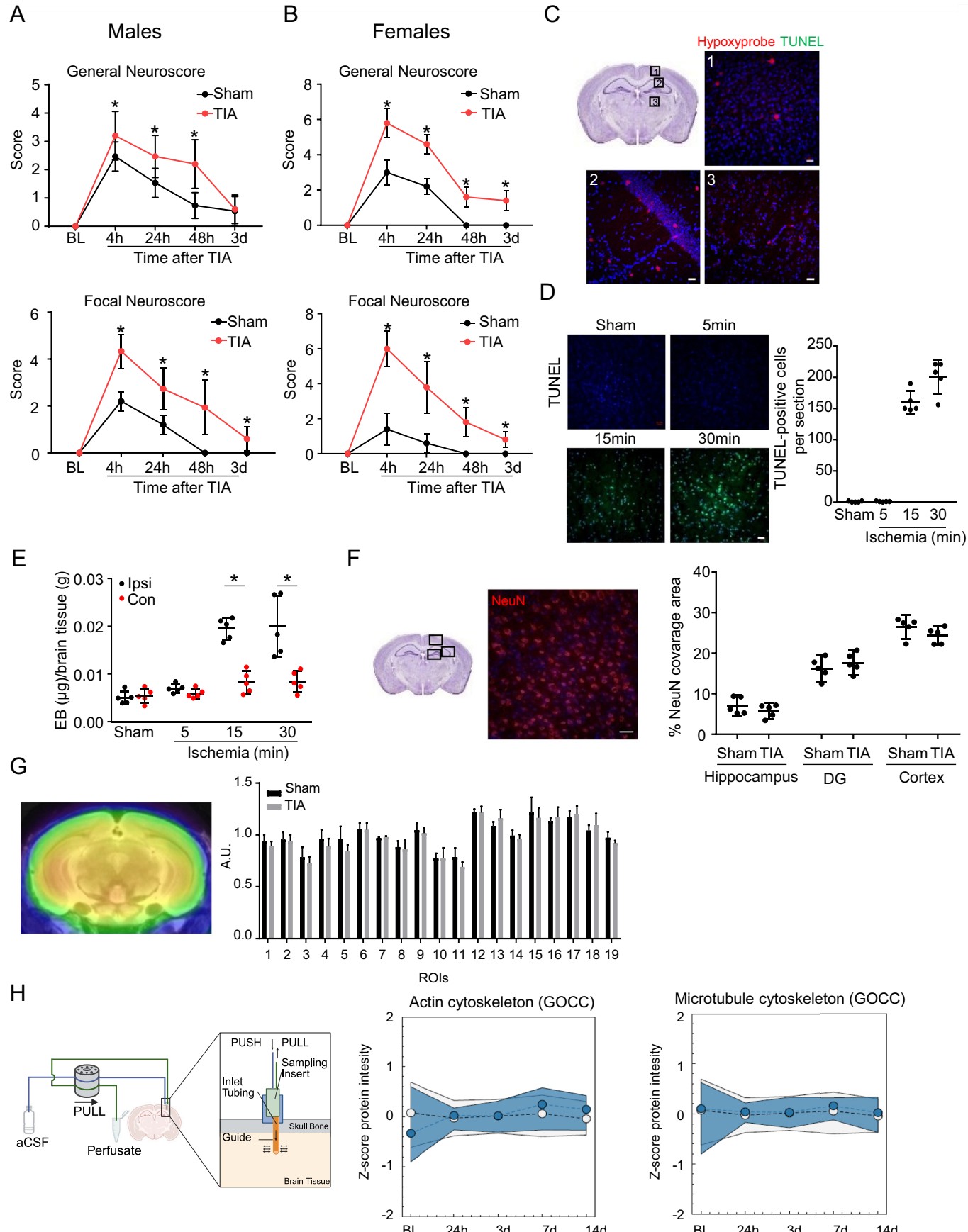

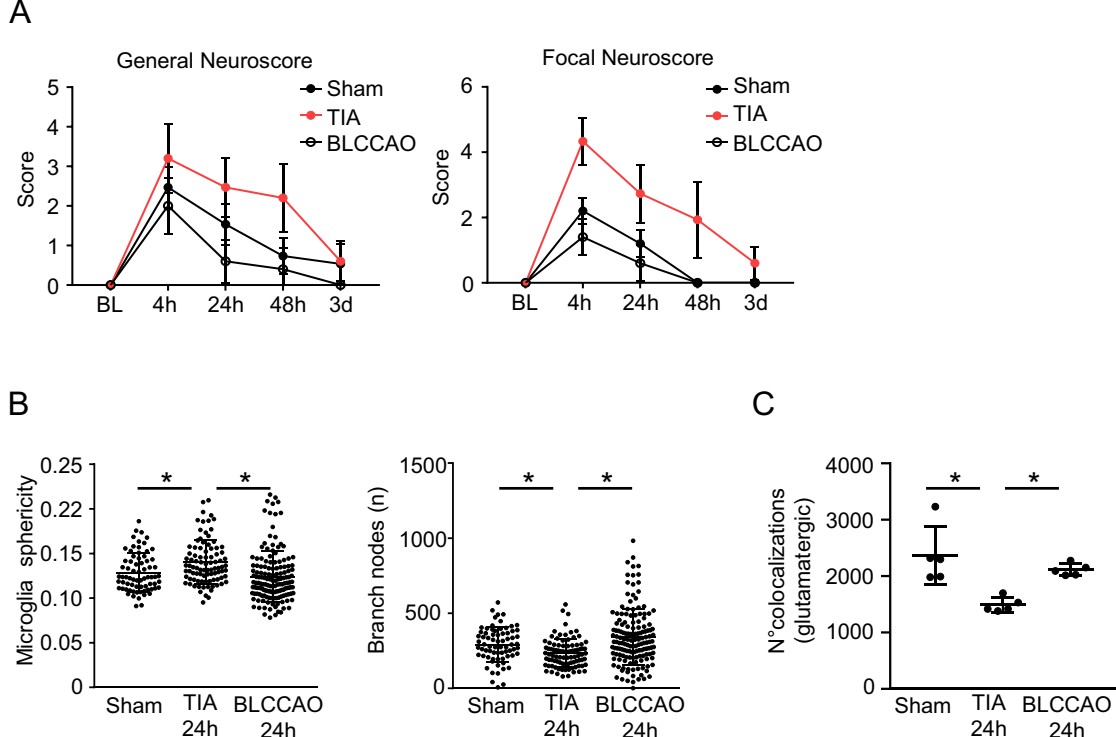

**Figure EV2. TIA vs BLCCAO.**

(A) General and focal Neuroscore at different time points after TIA, Sham and BLCCAO male animals ($n = 5$ per group). (B) Microglia morphology analysis (sphericity and branch nodes) at different time points 24 h after TIA, Sham and BLCCAO male animals ($n = 5$ per group; Microglia sphericity: TIA:$p$ value $= 0.0123$ and BLCCAO:$p$ value $<0.0001$), Branch nodes: TIA:$p$ value $= 0.0022$ and BLCCAO:$p$ value $<0.0001$. (C) Quantification of colocalized glutamatergic presynaptic and postsynaptic particles 24 h after TIA, Sham, and BLCCAO male animals ($n = 5$ per group; TIA:$p$ value $= 0.0140$ and BLCCAO:$p$ value $= 0.0400$). Statistical tests: (A–C) two-way ANOVA, corrected for multiple comparisons using two-stage step-up method of Benjamin Kriegel. Error bars indicate ±SD. *$p < 0.05$.

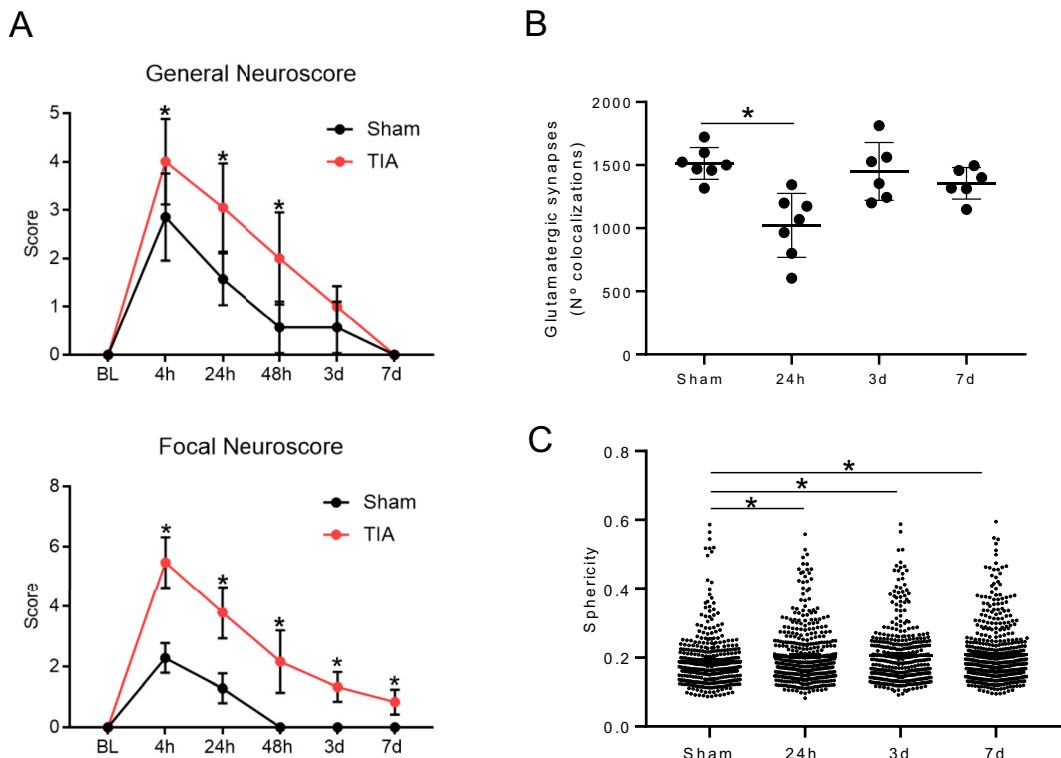

**Figure EV3.   TIA in aging.**

(A) General and focal Neuroscore at different time points after TIA in male aging animals (20 months) (Sham: $n = 7$; TIA: BL-24 h $n = 19$, 48 h-3d $n = 12$, 7 d $n = 6$; General neuroscore: 4 h: $p$ value = 0.0002, 24 and 48 h:$p$ value <0.0001; Focal Neuroscore: 4, 24, and 48 h:$p$ value <0.0001, 3 d:$p$ value = 0.0034). (B) Quantification of colocalized glutamatergic presynaptic and postsynaptic particles at different time points after TIA in male aging animals (20 months) (Sham: $n = 7$; TIA: 24 h $n = 7$, 3d-7d $n = 6$; 24 h:$p$ value = 0.0003). (C) Microglia morphology analysis (sphericity) at different time points after TIA in male aging animals (20 months) (Sham: $n = 7$; TIA: 24 h $n = 7$, 3d-7d $n = 6$; 24 h:$p$ value = 0.0083, 3 d:$p$ value = 0.0027, 7 d:$p$ value = 0.0011). Statistical tests: (A–C) two-way ANOVA, corrected for multiple comparisons using two-stage step-up method of Benjamin Kriegel. Error bars indicate ±SD. *$p < 0.05$.

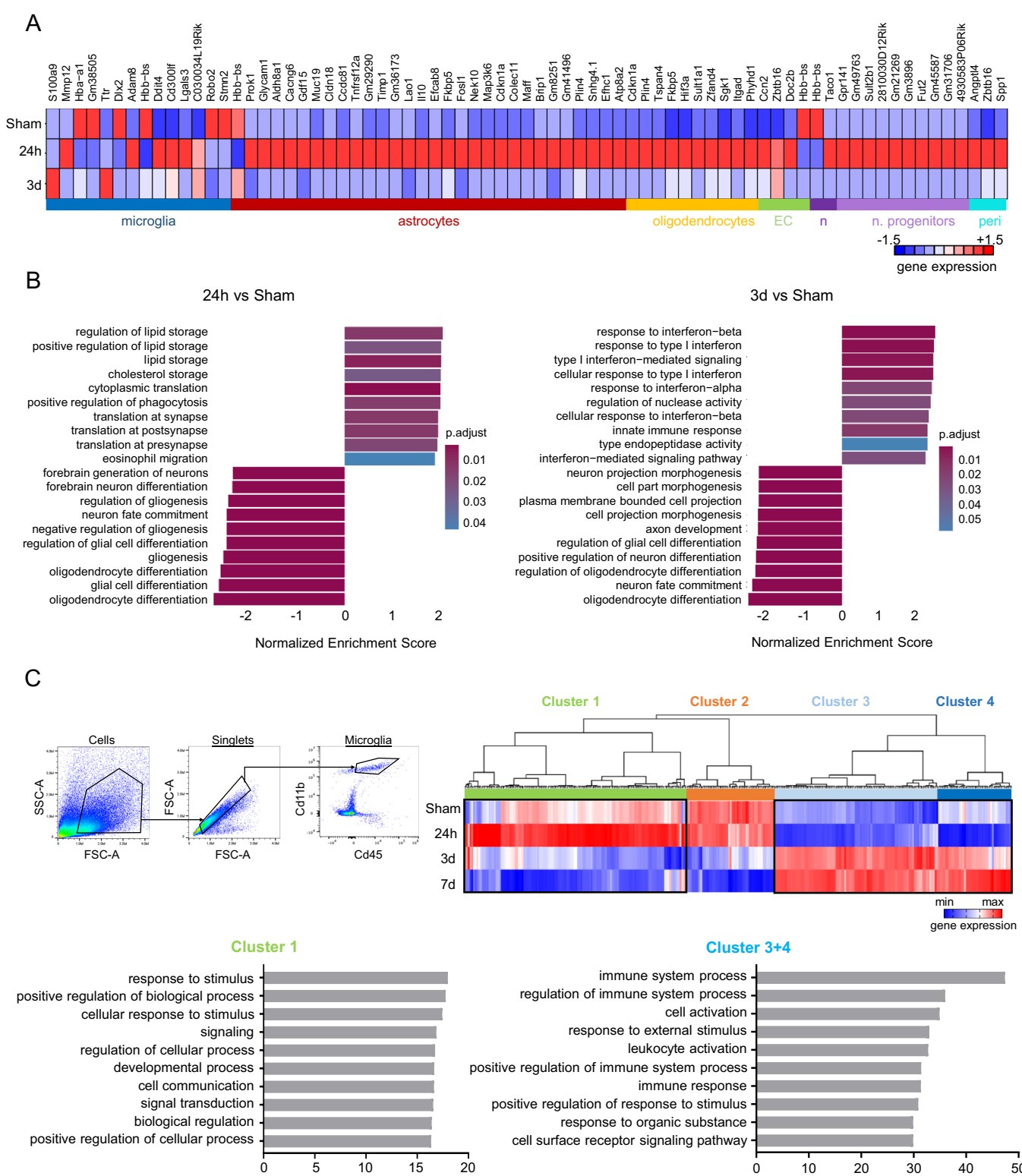

◀ **Figure EV4. The impact of TIA on microglia transcriptome.**

(A) Heat map of normalized and scaled (z-score) gene expression of significantly differentially up- and down-regulated genes in each cell type at 24 h and 3 d after TIA compared to Sham in males ($n = 3$ per group). (B) Gene set enrichment analyses (Biological Processes) of differentially regulated genes in microglial cells at 24 h and 3 d compared to Sham in males ($n = 3$ per group). (C) Representative FACS plot sort strategy for microglia-like cells isolation, cluster heat map showing ANOVA+ genes from Nanostring analysis from isolated microglia-like cells at different time points after TIA and pathway analysis from selected cluster 1 ($n = 3$ per group) and cluster 3 + 4 ($n = 3$ per group) in males. Statistical tests: (A) Wilcoxon rank sum test + bonferroni correction. (B) Genes ranked by log2 fold change, permutation-based GSEA, Benjamini–Hochberg correction, (C) Data were analyzed by ROSALIND® (https://rosalind.bio/). Significantly regulated genes were identified by ANOVA multiple sample testing (S0 = 0.1, permutation-based FDR = 0.05)

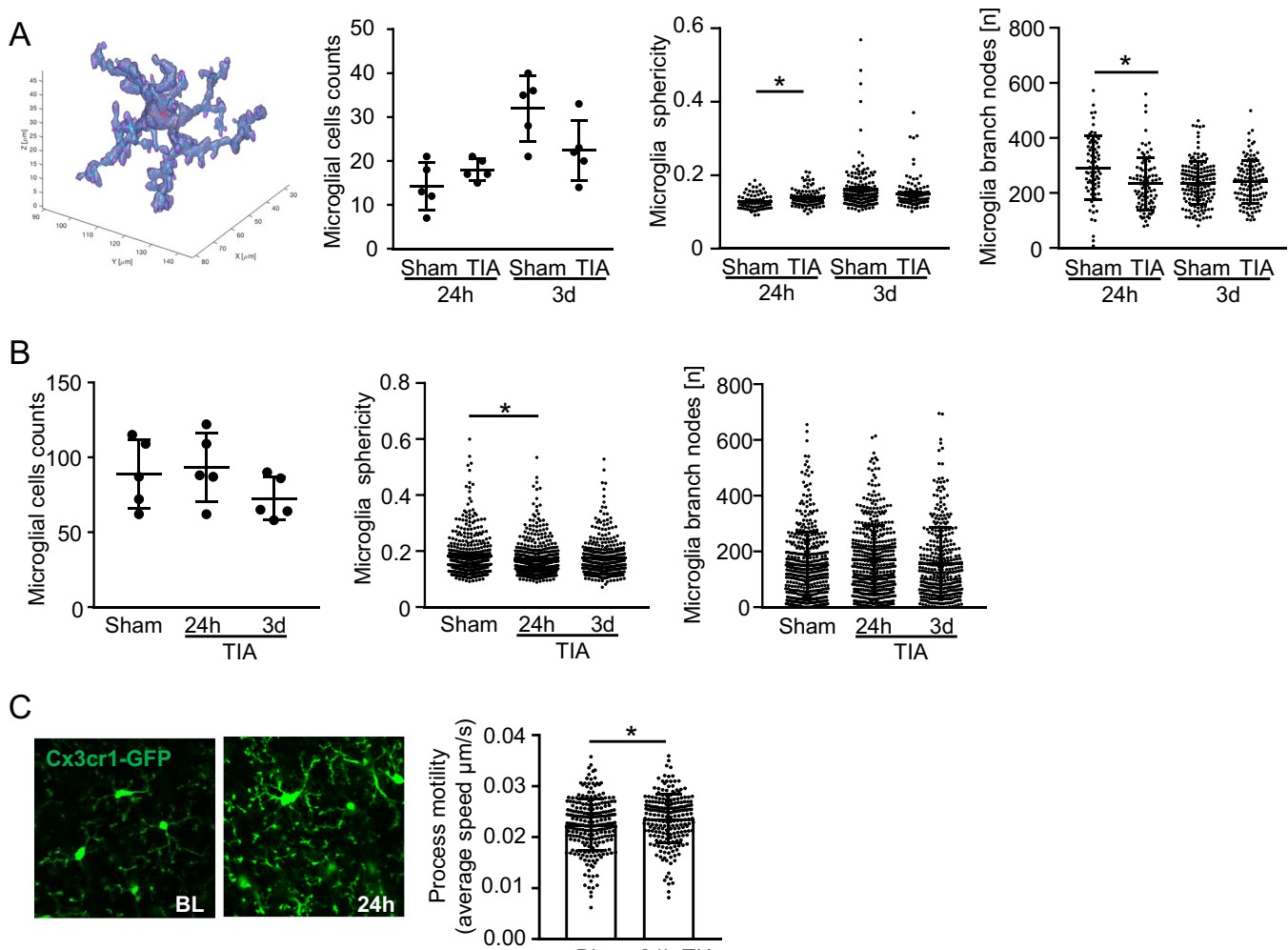

**Figure EV5. The impact of TIA on microglia morphology.**

(A) Representative image of 3D reconstructed microglia for microglia cells counts and microglia morphology analysis for two representative features: sphericity and branches nodes at 24 h after TIA ($n = 5$ per group; sphericity 24 h:$p$ value = 0.019; branch nodes 24 h:$p$ value = 0.022) in males and (B) females ($n = 5$ per group; sphericity 24 h:$p$ value = 0.0032). (C) Representative images of in vivo two-photon imaging of the microglia process motility and quantification at baseline (BL) and 24 h after TIA in males ($n = 8$ per group; 24 h:$p$ value = 0.025). Statistical tests: (A, C) two-way ANOVA, corrected for multiple comparisons using two-stage step-up method of Benjamin Kriegel. (B) two-way Student's $t$-test. Error bars are mean ± SD. *$P < 0.05$. Source data are available online for this figure.

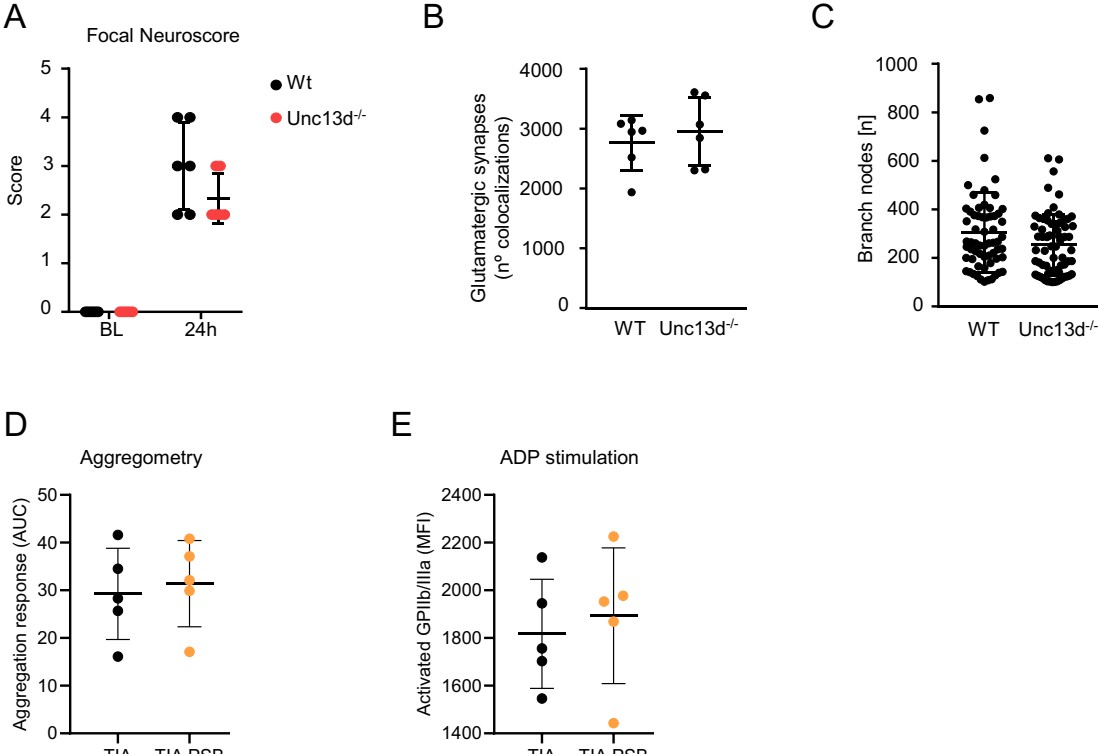

**Figure EV6. Platelets are not critical after a TIA.**

(A) Focal neuroscore, (B) glutamatergic synapses, and (C) microglia branch nodes analysis of Unc13KO and littermates wild-type (WT) animals 24 h after TIA in males. (D) Platelet aggregation and (E) activation in TIA in vehicle (Veh) or P2Y12Ri (-inhibitor) treated male animals ($n = 5$ per group). Statistical tests: (A–E) two-way Student's $t$-test. Error bars are mean ± SD.

