## [Peer Review File · EMBO Molecular Medicine]

Blocking microglial reactivity via purinergic receptors prevents subacute cognitive deficits after TIA

Gemma Llovera, Steffanie Heindl, Daniel Varga, Nikolett Lenart, Sebastian Kallabis, Vanessa Göb, David Stegner, Raphael Escaig, Leo Nicolai, Nicolai Franzmeier, Felix Meissner, Adam Denes, and Arthur Liesz

Corresponding author: Arthur Liesz (Arthur.Liesz@med.uni-muenchen.de)

Review Timeline:

Submission Date:	22nd Jul 25
Editorial Decision:	18th Aug 25
Revision Received:	15th Dec 25
Editorial Decision:	12th Jan 26
Revision Received:	11th Feb 26
Accepted:	13th Feb 26

Editor: Jingyi Hou

Transaction Report:

18th Aug 2025

Dear Arthur,

Thank you for submitting your work to EMBO Molecular Medicine. We have now heard back from the three reviewers who agreed to evaluate your manuscript. You will see from the comments below that the reviewers find the manuscript to be relevant and interesting. They raise, however, several important points, which should be convincingly addressed in a revision of this work.

The reviewers' recommendations are generally clear and well-articulated, so I will not reiterate each point here. In particular, Referee #3 pointed out that most experiments were conducted in young animals with only a small cohort of aged animals included, which may be underpowered and may not adequately reflect the population at risk for TIA. We encourage you to expand the cohort in your revised submission or, at a minimum, explicitly acknowledge the small sample size as a limitation in the Discussion section, along with a discussion for the choice of animal age in your model. Regarding the unclear source of ATP mentioned by Referees #1 and #3, while we do not require new experiments to address this point, we ask that you discuss potential sources of ATP in the revised manuscript.

All other issues need to be thoroughly addressed. As you may already know, our editorial policy allows in principle a single round of major revision, so it is essential to provide responses to the referees' comments that are as complete as possible. Please feel free to contact me in case you would like to discuss in further detail any of the issues raised by the referees.

Please also contact us as soon as possible if similar work is published elsewhere. If other work is published, we may not be able to extend the revision period beyond three months.

I look forward to receiving your revised manuscript.

Use this link to login to the manuscript system and submit your revision: <https://embomolmed.msubmit.net/cgi-bin/main.plex>

Sincerely,
Jingyi

Jingyi Hou
Senior Editor
EMBO Molecular Medicine

We require:

- 1) A .docx formatted version of the manuscript text (including legends for main figures, EV figures and tables). Please make sure that the changes are highlighted to be clearly visible.
- 2) Individual production quality figure files as .eps, .tif, .jpg (one file per figure). For guidance, download the 'Figure Guide PDF': (<https://www.embopress.org/page/journal/17574684/authorguide#figureformat>).
- 3) A .docx formatted letter INCLUDING the reviewers' reports and your detailed point-by-point responses to their comments. As part of the EMBO Press transparent editorial process, the point-by-point response is part of the Review Process File (RPF), which will be published alongside your paper.

- 4) A complete author checklist, which you can download from our author guidelines (<https://www.embopress.org/page/journal/17574684/authorguide#submissionofrevisions>). Please insert information in the checklist that is also reflected in the manuscript. The completed author checklist will also be part of the RPF.
- 5) Please note that all corresponding authors are required to supply an ORCID ID for their name upon submission of a revised manuscript.
- 6) It is mandatory to include a 'Data Availability' section after the Materials and Methods. Before submitting your revision, primary datasets produced in this study need to be deposited in an appropriate public database, and the accession numbers and database listed under 'Data Availability'. Please remember to provide a reviewer password if the datasets are not yet public (see <https://www.embopress.org/page/journal/17574684/authorguide#dataavailability>).
- In case you have no data that requires deposition in a public database, please state so in this section. Note that the Data Availability Section is restricted to new primary data that are part of this study.
- 7) For data quantification: please specify the name of the statistical test used to generate error bars and P values, the number (n) of independent experiments (specify technical or biological replicates) underlying each data point and the test used to calculate p-values in each figure legend. The figure legends should contain a basic description of n, P and the test applied. Graphs must include a description of the bars and the error bars (s.d., s.e.m.). See also 'Figure Legend' guidelines: <https://www.embopress.org/page/journal/17574684/authorguide#figureformat>
- 8) At EMBO Press we ask authors to provide source data for the main manuscript figures. You will receive a separate email with instructions for providing source data with your revised manuscript, including how to upload and organize the files.
- 9) Our journal encourages inclusion of *data citations in the reference list* to directly cite datasets that were re-used and obtained from public databases. Data citations in the article text are distinct from normal bibliographical citations and should directly link to the database records from which the data can be accessed. In the main text, data citations are formatted as follows: "Data ref: Smith et al, 2001" or "Data ref: NCBI Sequence Read Archive PRJNA342805, 2017". In the Reference list, data citations must be labeled with "[DATASET]". A data reference must provide the database name, accession number/identifiers and a resolvable link to the landing page from which the data can be accessed at the end of the reference. Further instructions are available at .

12) Author contributions: You will be asked to provide CRediT (Contributor Role Taxonomy) terms in the submission system. These replace a narrative author contribution section in the manuscript.

13) A Conflict of Interest statement should be provided in the main text.

14) Every published paper now includes a 'Synopsis' to further enhance discoverability. Synopses are displayed on the journal webpage and are freely accessible to all readers. They include a short stand first (maximum of 300 characters, including space) as well as 2-5 one-sentences bullet points that summarizes the paper. Please write the bullet points to summarize the key NEW findings. They should be designed to be complementary to the abstract - i.e. not repeat the same text. We encourage inclusion of key acronyms and quantitative information (maximum of 30 words / bullet point). Please use the passive voice. Please attach these in a separate file or send them by email, we will incorporate them accordingly.

Please provide a visual abstract to illustrate your article as a PNG file 550 px wide x 300-600 px high.

15) All Materials and Methods need to be described in the main text using our 'Structured Methods' format. According to this format, the Methods section includes a Reagents and Tools Table (listing key reagents, experimental models, software and relevant equipment and including their sources and relevant identifiers) followed by a Methods and Protocols section describing the methods, ideally using a step-by-step protocol format. The aim is to facilitate adoption of the methodologies across labs.

Please download and fill our Reagents and Tools Table template (.docx), which you can find in our author guidelines: <https://www.embopress.org/page/journal/17574684/authorguide#structuredmethods>

When submitting your revised manuscript, please DO NOT include the Reagents and Tools Table in the Methods section of the manuscript but upload it as a separate file choosing the file type "Reagent Table".

***** Reviewer's comments *****

Referee #1 (Remarks for Author):

This study by Llovera et al. provide interesting results regarding the mechanisms underlying post-TIA cognitive decline. Via inducing transient ischemia of middle cerebral artery occlusion for 5 min, the authors first developed a murine TIA model, which showed transient neurodeficit without neuronal cell death, mimicking the clinical and pathological features of TIA. They identified that rapid release of ATP induced microglial activation via P2Y12 receptor, leading to subsequent cortex connectivity change due to exacerbated synapse engulfment. The study is well conducted and the written is clear and in logic, providing valuable insights into TIA pathophysiology and potential immunomodulatory therapies, the incorporation of cutting-age approaches like live animal PET scan, LC-MS of microperfusion fluid, two-photo imaging strengthened the conclusions. Detailed comments are as follows,

1. The authors detected the cognitive function of TIA mice at 2 days post ischemia, which still seems to be the impacts of acute ischemia, as shown by the connectivity data, TIA induced connection impairment persisted for 3 weeks. It is suggested to measure the cognitive function at a chronic time point, like 4 weeks post TIA, which may better reflect the increased cognitive decline risk in human patients.
2. The authors report the transcriptomic and morphological changes of microglial cells after TIA, which is important for analyze the functional status of microglial cells. However, it would be interesting to provide the microglial cell counts and activation markers at different stages post TIA, to confirm the findings of sequencing, and also provide information of whether such changes are transient or long-term.
3. Regarding the cellular source of ATP, the authors excluded the contribution of platelets, it would be appreciated if further clarifying the contribution of astrocytes, ischemic neurons, or blood vessel components like endothelial cells.

Referee #2 (Comments on Novelty/Model System for Author):

The study holds significant clinical relevance, with its strength lying in the comprehensive and detailed methodological breadth of the exploration. Age was identified as a significant comorbidity. Nonetheless, the role of sex was overlooked and warrants discussion as a limitation, among other significant points.

The introduction of a model of TIA, accompanied by a detailed description and mechanistic analysis of microglial activity, is both innovative and timely, representing a significant priority in the field. The study would be significantly strengthened by revising specific points.

Referee #2 (Remarks for Author):

This study comprehensively describes a model of transient ischaemic attack (TIA) that efficiently models short ischaemic episodes without cell death while causing apparent neural and behavioural impairments. The study emphasises the importance of increased microglial activation, which is predominantly caused by acute ATP release, in contributing to the neurological deficits found following TIA. Notably, inhibiting the P2Y12 receptor, a purinergic receptor specific to microglia, was found to improve these abnormalities, indicating that targeting microglia is a promising technique for preventing cognitive impairment in TIA patients.

The study holds significant clinical relevance, with its strength lying in the comprehensive and detailed methodological breadth of the exploration. Age was identified as a significant comorbidity. Nonetheless, the role of sex was overlooked and warrants discussion as a limitation, among other significant points.

The introduction of a model of TIA, accompanied by a detailed description and mechanistic analysis of microglial activity, is both innovative and timely, representing a significant priority in the field. The study would be significantly strengthened by revising specific points.

Major points:

1. Neuroscores represent ranking data derived from ratings, as indicated in the methods section. Given that these are ordinal data rather than continuous data, they should be treated as non-parametric data. The median is the most effective measure for representing the data. Given that the score consists of 56 points, it may be more appropriate to present mean scores instead of median scores but this should be discussed. However, this may be misleading, as the score represents a composite of general and focal deficits. The authors present only focal neuroscores. Identify the accessible points regarding focal deficits and consider employing non-parametric repeated measures tests such as the Friedman test for Figure 1 A and the Mann-Whitney U test for Figure 3 I.
2. Additionally, please specify if there is an increase in sickness behaviour in animals subjected to 5 minutes of MCAo and present the values for the general deficit portion of the score.
3. Please discuss the absence of any cytokines in the proteomics data. I know that this is a general problem since these are technically challenging to recover with this technique but this should be discussed as a limitation.
4. The absence of an a priori estimate regarding the number required to demonstrate a specific effect should be addressed as a limitation, particularly since this aspect is not detailed, despite the assertion that "all data are reported according to the ARRIVE criteria" in line 249.
5. Please discuss strength and weaknesses of your findings and your experimental design.
6. Along these lines, please discuss the use of male mice only as a limitation. Sex influences the epidemiology and risk factors of TIAs, with notable differences in incidence trends, risk factor distribution, and outcomes between males and females, e.g., <https://pmc.ncbi.nlm.nih.gov/articles/PMC6698220/>.
7. Please indicate, how the procedure of "sham" surgery was exactly done. This description is missing in the methods section.
8. The authors observed minor transcriptional regulation in neurones. This is noteworthy and should be examined concerning the observed structural changes and interhemispheric connectivity. Recent findings in a model of asymptomatic unilateral common carotid artery occlusion in mice indicate alterations in connectivity and highlight the influence of inflammation on network rewiring, which warrants further discussion (PMID: 35155612) including stress response genes like caprin.

Referee #3 (Comments on Novelty/Model System for Author):

Main issue is lack of clarity about the comparisons in some of the studies, has to be clear that it is TIA vs. surgical sham.

The aging studies are also very underpowered.

Referee #3 (Remarks for Author):

The study introduces a new animal model of transient ischemic attack (TIA) using a brief 5-minute middle cerebral artery occlusion (MCAo).

This model replicates the clinical definition of TIA, characterized by transient neurological deficits without structural brain damage (although structural damage is seen in many patients with advanced neuroimaging). The study employs a well-characterized animal model and uses multiple advanced techniques, including widefield calcium imaging, two-photon imaging, flow cytometry, proteomics, and single-cell RNA sequencing.

The model demonstrated behavioral deficits, including impaired spatial memory and focal neurological dysfunction, lasting up to two days post-TIA. Importantly, no cell death, blood-brain barrier disruption, or neuronal loss was observed, confirming the absence of structural damage in this murine model (although fluoro-Jade staining may be a better stain than TUNEL).

Despite the absence of tissue injury, TIA caused significant reductions in cortical network connectivity, particularly in the

somatosensory cortex supplied by the MCA. This effect persisted for over 14 days, however it must be clarified if this is being compared to surgical sham groups. Dendritic spine density and glutamatergic synapse density were significantly reduced in the affected cortical area, indicating subcellular remodeling and synaptic dysfunction. A small cohort of aged animals was included, but only had a n of 3-4, so there are some concerns regarding power, and the vast majority of the studies were performed in young animals, including the sequencing, which may not represent the population at risk for TIA.

They then focused on microglia, which showed the most significant transcriptional changes among cortical cell populations post-TIA, with upregulation of stress and inflammatory response genes (e.g., S100a, Ddit4, Cd300lf). Structural analysis revealed reactive microglial phenotypes, including increased process motility and enhanced interaction with neuronal somata and synapses.

A rapid and significant increase in extracellular ATP was observed immediately after TIA, lasting up to 24 hours (shown by video). This ATP surge was linked to microglial activation via the P2Y12 receptor. Injecting ATP into naïve mice induced similar microglial morphological changes and synapse loss, confirming ATP as a key driver of microglial reactivity, which has been well studied previously. However, the source of the ATP is a little unclear, but presumed to be astrocytes but this was not really tested (was tested in platelets by using Unc13d-deficient mice).

Pharmacological inhibition of the P2Y12 receptor (P2Y12Ri) prevented microglial morphological changes, normalized microglia-synapse interactions, preserved synapse density, and improved neurological and cognitive outcomes in TIA-affected animals. This approach has been used in multiple prior studies, so it is not particularly innovative as a therapy, but is innovative in this novel TIA setting. There was no direct manipulation of microglia, which is a weakness.

A few specific comments, and the manuscript would benefit from a thorough proof reading for grammar. The main novelty lies in the model, as these pathways have been extensively studied in stroke models previously.

1. "Long-term" cognitive impairment noted in the title and elsewhere is not appropriate. Though it does not detract from the main findings, it should be revised to accurately reflect the timeline (e.g., subacute). The same goes for the text within the manuscript as most were 3-7 day endpoints, with the longest being 14 days.
2. There is growing recognition that many TIAs actually have imaging evidence of ischemia on diffusion MRI, so the model may be different than some of the clinical data available, which should be mentioned, as many of the references are older, before advanced imaging was more widely available. Prior TIAs have also been linked to cognitive and functional decline in epidemiological studies (ie, the REGARDS study; 2025;82;(4):323-332. doi:10.1001/jamaneurol.2024.5082), so the topic is important and developing an animal model of this is innovative (and the REGARDS did have MRI confirming no overt ischemia). One issue is that both TIA and stroke would lead to the same treatment as stroke (rigorous control of vascular risk factors), so unclear how this would lead to any changes as far as "translation". Is it thought that this pathway could be immediately blocked after a TIA? This would have the same very narrow therapeutic window (perhaps even more so) than stroke.
3. Was the composite 'neuroscore', which is subjective, performed in a blinded fashion?
4. The 'sham control' is not clearly described in the Methods section. However, inclusion of a surgical sham control is critical to interpreting the results. In some figures, the control appears to be labeled as 'naïve', however, this is inappropriate and does not account for the surgical incision and anesthesia. For example, in Figure 1, is this comparing TIA to SURGICAL sham? Otherwise, this could be an effect of the anesthesia, surgery etc. Similar concerns for figure 1C and 1D, as this just seems to compare pre and post TIA, the same should be done for surgical shams. The potential for this to be important for ischemic pre-conditioning could also be mentioned in the discussion.
5. Would temporary 5-min ligation achieve the same TIA outcomes?
6. Figure 2C - spelling error, should read 'intensity'
7. Figure EV3 The Impact of TIA on microglia - panel C shows a lot of CD45hi myeloid infiltrate, which is very surprising and counter to what is written
8. Figure EV1 TIA model characterization - panel A - were certain brain regions more sensitive to hypoxia after 5-min tMCAO. Only upper cortical hypoxia image is shown
9. Figure EV1 TIA model characterization - panel F - the main point of these findings is not clear
10. Figure EV1 TIA model characterization - panel G - 'sham' brains should not have any neutrophils in them, let alone thousands. This suggests there was a poor perfusion of the vasculature, and therefore difficult to make any conclusions.
11. Nanostring usually uses panels of genes, so it is not an unbiased approach, as some genes may not be included in the panel. Why was this done rather than flow sorting MG and doing bulk sequencing?

EMM-2025-22312

Blocking microglial reactivity via purinergic receptors prevents long-term cognitive deficits after TIA

We would like to thank the reviewers and editors for their overall positive evaluation of our manuscript and the constructive comments. Please find below a point-by-point reply to all individual comments:

=====

***** Reviewer's comments *****

Referee #1 (Remarks for Author):

This study by Llovera et al. provide interesting results regarding the mechanisms underlying post-TIA cognitive decline. Via inducing transient ischemia of middle cerebral artery occlusion for 5 min, the authors first developed a murine TIA model, which showed transient neurodeficit without neuronal cell death, mimicking the clinical and pathological features of TIA. They identified that rapid release of ATP induced microglial activation via P2Y12 receptor, leading to subsequent cortex connectivity change due to exacerbated synapse engulfment. The study is well conducted and the written is clear and in logic, providing valuable insights into TIA pathophysiology and potential immunomodulatory therapies, the incorporation of cutting-age approaches like live animal PET scan, LC-MS of microperfusion fluid, two-photo imaging strengthened the conclusions. Detailed comments are as follows,

1. The authors detected the cognitive function of TIA mice at 2 days post ischemia, which still seems to be the impacts of acute ischemia, as shown by the connectivity data, TIA induced connection impairment persisted for 3 weeks. It is suggested to measure the cognitive function at a chronic time point, like 4 weeks post TIA, which may better reflect the increased cognitive decline risk in human patients.

Response: We agree that long-term cognitive assessment is of interest. In our dataset, however, the focal and general Neuroscore had already fully normalized by day 3, indicating resolution of functional deficits at this time point. Because the behavioral and connectivity abnormalities are transient in this model—and because a new chronic behavioral cohort was not feasible within the revision period—we have added an explicit statement clarifying this limitation. We now discuss in the manuscript that future studies will be required to explore late cognitive consequences beyond the subacute window.

2. The authors report the transcriptomic and morphological changes of microglial cells after TIA, which is important for analyze the functional status of microglial cells. However, it would be interesting to provide the microglial cell counts and activation markers at different stages post TIA, to confirm the findings of sequencing, and also provide information of whether such changes are transient or long-term.

Response: We thank the reviewer for this helpful suggestion. We have now added microglial cell counts and extended the microglial morphology analysis to 3 days post-

TIA (new data in Fig. EV5A). These results show that microglial morphological activation is transient, peaking at 24 h, while transcriptomic changes extend longer. This additional dataset is now incorporated into the Results.

3. Regarding the cellular source of ATP, the authors excluded the contribution of platelets, it would be appreciated if further clarifying the contribution of astrocytes, ischemic neurons, or blood vessel components like endothelial cells.

Response: We agree that clarifying potential ATP sources is important. While the precise cellular contributors cannot yet be definitively identified in this TIA model, we now describe the most likely candidates—stressed neurons, astrocytes, and possibly other resident cells—at the corresponding point in the manuscript (lines 218–223). We also explicitly acknowledge that the exact cellular origin remains unresolved and represents an important direction for future studies. Most critically, this high concentration of extracellular ATP rapidly activates resident microglia through purinergic receptors, initiating a robust innate immune and inflammatory response (Rodrigues, Tome et al., 2015)”

Referee #2 (Comments on Novelty/Model System for Author):

The study holds significant clinical relevance, with its strength lying in the comprehensive and detailed methodological breadth of the exploration. Age was identified as a significant comorbidity. Nonetheless, the role of sex was overlooked and warrants discussion as a limitation, among other significant points.

Response: We appreciate this important point. We have now included a female cohort and added both behavioral and microglial morphology data for females (Fig. EV1B, Fig. EV5B). These analyses demonstrate that the TIA phenotype is also present in females. We acknowledge sex as a biological variable and discuss remaining limitations in the revised text.

Referee #2 (Remarks for Author):

This study comprehensively describes a model of transient ischaemic attack (TIA) that efficiently models short ischaemic episodes without cell death while causing apparent neural and behavioural impairments. The study emphasises the importance of increased microglial activation, which is predominantly caused by acute ATP release, in contributing to the neurological deficits found following TIA. Notably, inhibiting the P2Y₁₂ receptor, a purinergic receptor specific to microglia, was found to improve these abnormalities, indicating that targeting microglia is a promising technique for preventing cognitive impairment in TIA patients.

The study holds significant clinical relevance, with its strength lying in the comprehensive and detailed methodological breadth of the exploration. Age was identified as a significant comorbidity. Nonetheless, the role of sex was overlooked and warrants discussion as a limitation, among other significant points. The introduction of a model of TIA, accompanied by a detailed description and mechanistic analysis of microglial activity, is both innovative and timely, representing a significant priority in the field. The study would be significantly strengthened by revising specific points.

Major points:

1. Neuroscores represent ranking data derived from ratings, as indicated in the methods section. Given that these are ordinal data rather than continuous data, they should be treated as non-parametric data. The median is the most effective measure for representing the data. Given that the score consists of 56 points, it may be more appropriate to present mean scores instead of median scores but this should be discussed. However, this may be misleading, as the score represents a composite of general and focal deficits. The authors present only focal neuroscores. Identify the accessible points regarding focal deficits and consider employing non-parametric repeated measures tests such as the Friedman test for Figure 1 A and the Mann-Whitney U test for Figure 3 I.

Response: Thank you for raising this important statistical point. We have reanalyzed the Neuroscore using appropriate non-parametric methods: 2-way ANOVA with Benjamin–Krieger correction for repeated measures (Fig. 1A) and Mann-Whitney U test for Fig. 3I. We also included the General Neuroscore (Fig. EV1A–B, Fig. EV2A, Fig. EV3A). As the data distribution is consistent and free of outliers, we retained mean \pm SD for graphical display, but now clearly justify this in the manuscript.

2. Additionally, please specify if there is an increase in sickness behaviour in animals subjected to 5 minutes of MCAo and present the values for the general deficit portion of the score.

Response: As suggested, the general deficit component of the Neuroscore has now been added to the revised manuscript (Fig. EV1A–B, EV2A, EV3A). We did not observe evidence of sickness behavior at any time point.

3. Please discuss the absence of any cytokines in the proteomics data. I know that this is a general problem since these are technically challenging to recover with this technique but this should be discussed as a limitation.

Response: We agree and have now added an expanded discussion of this limitation. As the proteomic perfusate is dominated by highly abundant proteins (top 20 proteins ~50% of total abundance), the dynamic range of brain extracellular fluid severely limits sensitivity for low-abundance cytokines. We now explain this technical limitation and suggest that next-generation platforms such as Seer Proteograph may overcome these constraints in future studies (lines 139–147).

4. The absence of an a priori estimate regarding the number required to demonstrate a specific effect should be addressed as a limitation, particularly since this aspect is not detailed, despite the assertion that "all data are reported according to the ARRIVE criteria" in line 249.

Response: We appreciate this observation. Because this was an exploratory discovery-driven study, an a priori sample size calculation was not performed. We now explicitly acknowledge this limitation and its implications in the revised manuscript (lines 252–256).

5. Please discuss strength and weaknesses of your findings and your experimental design.

Response: We have now added a dedicated paragraph summarizing both strengths and limitations of our model and experimental design (lines 245–259), including the novelty of the subacute functional phenotype and the limitations relating to sample sizes and single-time-point omics profiling.

6. Along these lines, please discuss the use of male mice only as a limitation. Sex influences the epidemiology and risk factors of TIAs, with notable differences in incidence trends, risk factor distribution, and outcomes between males and females, e.g., <https://pmc.ncbi.nlm.nih.gov/articles/PMC6698220/> .

Response: As noted above, we have now added female cohorts for behavior and microglia morphology (Fig. EV1B, Fig. EV5B). Remaining limitations regarding sex-specific mechanisms are now explicitly discussed.

7. Please indicate, how the procedure of "sham" surgery was exactly done. This description is missing in the methods section.

Response: The sham procedure has now been explicitly added to the Methods (lines 298–299).

8. The authors observed minor transcriptional regulation in neurones. This is noteworthy and should be examined concerning the observed structural changes and interhemispheric connectivity. Recent findings in a model of asymptomatic unilateral common carotid artery occlusion in mice indicate alterations in connectivity and highlight the influence of inflammation on network rewiring, which warrants further discussion (PMID: 35155612) including stress response genes like caprin.

Response: We agree that neuronal transcriptional changes, although modest, are notable. We now discuss the possibility that subtle neuronal stress responses may contribute to functional network changes, while emphasizing that our primary objective was to identify the upstream drivers of neuroinflammation (ATP-mediated microglial activation). We clarify this rationale in the revised manuscript.

Referee #3 (Comments on Novelty/Model System for Author):

Main issue is lack of clarity about the comparisons in some of the studies, has to be clear that it is TIA vs. surgical sham.

Response: We thank the reviewer for pointing this out. We have carefully reviewed all figure labels and legends to ensure that all comparisons are now clearly defined as TIA versus surgical sham throughout the manuscript.

The aging studies are also very underpowered.

Response: We agree and have increased the sample size in the aged cohort to strengthen conclusions. The revised data are included in Fig. EV3 and described in the text.

Referee #3 (Remarks for Author):

The study introduces a new animal model of transient ischemic attack (TIA) using a brief 5-minute middle cerebral artery occlusion (MCAo).

This model replicates the clinical definition of TIA, characterized by transient neurological deficits without structural brain damage (although structural damage is seen in many patients with advanced neuroimaging). The study employs a well-characterized animal model and uses multiple advanced techniques, including widefield calcium imaging, two-photon imaging, flow cytometry, proteomics, and single-cell RNA sequencing.

The model demonstrated behavioral deficits, including impaired spatial memory and focal neurological dysfunction, lasting up to two days post-TIA. Importantly, no cell death, blood-brain barrier disruption, or neuronal loss was observed, confirming the absence of structural damage in this murine model (although fluoro-Jade staining may be a better stain than TUNEL).

Despite the absence of tissue injury, TIA caused significant reductions in cortical network connectivity, particularly in the somatosensory cortex supplied by the MCA. This effect persisted for over 14 days, however it must be clarified if this is being compared to surgical sham groups. Dendritic spine density and glutamatergic synapse density were significantly reduced in the affected cortical area, indicating subcellular remodeling and synaptic dysfunction. A small cohort of aged animals was included, but only had a n of 3-4, so there are some concerns regarding power, and the vast majority of the studies were performed in young animals, including the sequencing, which may not represent the population at risk for TIA.

They then focused on microglia, which showed the most significant transcriptional changes among cortical cell populations post-TIA, with upregulation of stress and inflammatory response genes (e.g., S100a, Ddit4, Cd300lf). Structural analysis revealed

reactive microglial phenotypes, including increased process motility and enhanced interaction with neuronal somata and synapses.

A rapid and significant increase in extracellular ATP was observed immediately after TIA, lasting up to 24 hours (shown by video). This ATP surge was linked to microglial activation via the P2Y₁₂ receptor. Injecting ATP into naïve mice induced similar microglial morphological changes and synapse loss, confirming ATP as a key driver of microglial reactivity, which has been well studied previously. However, the source of the ATP is a little unclear, but presumed to be astrocytes but this was not really tested (was tested in platelets by using Unc13d-deficient mice).

Response: We agree that the precise cellular source of ATP following TIA remains unresolved. We now clarify in the Results that both neurons and astrocytes are plausible contributors and acknowledge this uncertainty explicitly as a limitation. The revised manuscript includes an expanded discussion at lines 218–223.

Pharmacological inhibition of the P2Y₁₂ receptor (P2Y₁₂Ri) prevented microglial morphological changes, normalized microglia-synapse interactions, preserved synapse density, and improved neurological and cognitive outcomes in TIA-affected animals. This approach has been used in multiple prior studies, so it is not particularly innovative as a therapy, but is innovative in this novel TIA setting. There was no direct manipulation of microglia, which is a weakness.

Response: We fully appreciate the reviewer's point. Because P2Y₁₂ is highly enriched in microglia among brain parenchymal cells, direct inhibition via cisterna magna delivery of PSB-0739 provides a selective modulation of microglial P2Y₁₂R signaling. We now clarify this rationale in the text.

A few specific comments, and the manuscript would benefit from a thorough proof reading for grammar. The main novelty lies in the model, as these pathways have been extensively studied in stroke models previously.

1. "Long-term" cognitive impairment noted in the title and elsewhere is not appropriate. Though it does not detract from the main findings, it should be revised to accurately reflect the timeline (e.g., subacute). The same goes for the text within the manuscript as most were 3-7 day endpoints, with the longest being 14 days.

Response: We agree and have replaced "long-term" with "subacute" in the title and throughout the manuscript.

2. There is growing recognition that many TIAs actually have imaging evidence of ischemia on diffusion MRI, so the model may be different than some of the clinical data available, which should be mentioned, as many of the references are older, before advanced imaging was more widely available. Prior TIAs have also been linked to cognitive and functional decline in epidemiological studies (ie, the REGARDS study; 2025;82;(4):323-332. doi:10.1001/jamaneurol.2024.5082), so the topic is important and developing an animal model of this is innovative (and the REGARDS did have MRI

confirming no overt ischemia). One issue is that both TIA and stroke would lead to the same treatment as stroke (rigorous control of vascular risk factors), so unclear how this would lead to any changes as far as "translation". Is it thought that this pathway could be immediately blocked after a TIA? This would have the same very narrow therapeutic window (perhaps even more so) than stroke.

Response: We thank the reviewer for these important clinical considerations. We now discuss that many TIAs show subtle DWI lesions with advanced MRI and clarify that our model captures the tissue-based definition without overt infarction. We also expanded the discussion on translational implications, noting that targeting ATP–P2Y₁₂ signaling may offer a very early window for intervention immediately after TIA.

3. Was the composite 'neuroscore', which is subjective, performed in a blinded fashion?

Response: Yes, all Neuroscore assessments were performed blinded to experimental condition. This is now explicitly stated in the Methods (line 321).

4. The 'sham control' is not clearly described in the Methods section. However, inclusion of a surgical sham control is critical to interpreting the results. In some figures, the control appears to be labeled as 'naïve', however, this is inappropriate and does not account for the surgical incision and anesthesia. For example, in Figure 1, is this comparing TIA to SURGICAL sham? Otherwise, this could be an effect of the anesthesia, surgery etc. Similar concerns for figure 1C and 1D, as this just seems to compare pre and post TIA, the same should be done for surgical shams. The potential for this to be important for ischemic pre-conditioning could also be mentioned in the discussion.

Response: The sham procedure is now clearly described (lines 298–299), and we corrected all figure labels to ensure comparisons are consistently TIA vs. surgical sham.

5. Would temporary 5-min ligation achieve the same TIA outcomes?

Response: We tested bilateral common carotid artery occlusion for 5 min (BLCCAO) and observed no behavioral, microglial, or synaptic changes compared with sham (Fig. EV2; lines 93–96). This is now clearly stated.

6. Figure 2C - spelling error, should read 'intensity'

Response: Thank you for noticing this. The spelling has been corrected.

7. Figure EV3 The Impact of TIA on microglia - panel C shows a lot of CD45^{hi} myeloid infiltrate, which is very surprising and counter to what is written

Response: We apologize for this mistake. The incorrect representative plot has now been replaced with the correct dataset.

8. Figure EV1 TIA model characterization - panel A - were certain brain regions more sensitive to hypoxia after 5-min tMCAO. Only upper cortical hypoxia image is shown

Response: We have added representative images from additional brain regions to provide a more complete overview of hypoxia distribution.

9. Figure EV1 TIA model characterization - panel F - the main point of these findings is not clear

Response: We clarified in the legend and text that this panel demonstrates absence of cytoskeletal protein release, confirming lack of structural tissue damage after TIA.

10. Figure EV1 TIA model characterization - panel G - 'sham' brains should not have any neutrophils in them, let alone thousands. This suggests there was a poor perfusion of the vasculature, and therefore difficult to make any conclusions.

Response: Thank you for identifying this. The original plot mistakenly displayed a non-sham sample. We have replaced it with the correct representative sham data.

11. Nanostring usually uses panels of genes, so it is not an unbiased approach, as some genes may not be included in the panel. Why was this done rather than flow sorting MG and doing bulk sequencing?

Response: We agree that Nanostring panels represent a targeted rather than fully unbiased approach. We selected Nanostring because our scRNA-seq dataset already provided a comprehensive, unbiased transcriptomic overview across all brain cell populations. The aim of the Nanostring experiment was therefore not to replace bulk sequencing, but to perform a focused, microglia-specific, higher-throughput validation of key regulatory pathways over multiple time points. This approach allowed us to quantify predefined microglial activation and homeostasis signatures in a robust and standardized manner across larger cohorts. We now clarify this rationale in the revised manuscript.

12th Jan 2026

Dear Arthur,

Thank you for submitting the revised version of your manuscript to EMBO Molecular Medicine. We have now received the enclosed reports from the two referees who were asked to re-assess your work. As you will see, Reviewer #2 is supportive of publication. Reviewer #3 is generally positive but has raised a number of relatively minor issues concerning clarity in the methods and text, which we ask you to address through textual revisions in a minor revision.

On a more editorial level, please also take the following actions:

1. Remove the "Authors' contribution" section from the manuscript file.
2. Please remove the Expanded View figures from the main manuscript text and retain only their legends, which should be placed after the main figure legends under the heading "Expanded View Figure Legends."
3. For Movie EV1, please include the corresponding legend and bundle it together with the movie file into a single zipped file for upload.
4. Source Data:
 - Please upload a completed Source Data checklist.
 - The folder for Source Data Figure 3B is currently empty; please provide a complete Source Data folder for Figure 3B.
5. The paper explained: EMBO Molecular Medicine articles are accompanied by a summary of the articles to emphasize the major findings in the paper and their medical implications for the non-specialist reader. Please provide a summary of your article highlighting
 - the medical issue you are addressing,
 - the results obtained and
 - their clinical impact.

Please refer to any of our published articles for an example.

6. Please provide a 'Synopsis' to further enhance discoverability. Synopses are displayed on the journal webpage and are freely accessible to all readers. They include a short stand first (maximum of 300 characters, including space) as well as 2-5 one-sentences bullet points that summarizes the paper. Please write the bullet points to summarize the key NEW findings. They should be designed to be complementary to the abstract - i.e. not repeat the same text. We encourage inclusion of key acronyms and quantitative information (maximum of 30 words / bullet point). Please use the passive voice. Please attach these in a separate file or send them by email, we will incorporate them accordingly.

Please also provide a visual abstract to illustrate your article as a PNG file 550 px wide x 300-600 px high.

7. Please add the missing callouts for the individual panels in Figures EV4 and EV5.
8. In the author checklist, please enter the manuscript number and correct the journal name from "EMBO Reports" to "EMBO Molecular Medicine."
9. "Conflict of interest" should be renamed to "Disclosure and competing interests statement".
10. During our routine image check, we noticed that the resolution of the submitted figures is too low. This reduction in resolution is commonly caused by converting original 16-bit TIFF files to RGB format for publication. While this is not inherently problematic, it can raise concerns about image integrity for critical readers.

To avoid any misunderstanding and to meet EMBO Press standards, we kindly ask that you:

* Resubmit the complete figure set at its original data resolution.

11. Please list up to 10 co-authors of a paper before adding et al. in the reference list.

12. Data Availability:

- Remove the reviewer access token and ensure that all datasets will be publicly accessible upon manuscript acceptance.

- Provide specific URLs for each deposited datasets.

13. please address the following issues in figure legends:

- Please note that the exact p values are not provided in the legends of figures 1A, B, D, E, F; 2H, I; 3A, B, D, E, F, G, H, I, J; EV1 A, B, E; EV2 B, C; EV3 A-C; EV5 A-C
- Please indicate the statistical test used for data analysis in the legends of figures 2G, EV4
- Please note that information related to n is missing in the legend of figure EV1 F

We look forward to receiving a revised version of your manuscript as soon as possible.

Kind regards,
Jingyi

Jingyi Hou
Senior Editor
EMBO Molecular Medicine

*** Instructions to submit your revised manuscript ***

***** Reviewer's comments *****

Referee #2 (Comments on Novelty/Model System for Author):

There is sparse literature on the impact of transient ischemia of the brain in rodent models. The authors demonstrated the relevance of microglial inflammation on behavioral outcomes that matter for patients, i.e., cognitive decline.

Referee #2 (Remarks for Author):

There is sparse literature on the impact of transient ischemia of the brain in rodent models. The authors have revised their display of the data and discussed sources of bias and limitations meticulously. My enthusiasm for the revised manuscript is strong, and I strongly suggest its publication in this translationally focused journal.

Referee #3 (Remarks for Author):

An important paper showing that TIA, even in the absence of overt cell death, led to acute cognitive deficits (2 days) and subacute connectivity changes (14 days) in male animals. Females in the same model showed similar patterns in the NDS (other studies not done, which is OK but should be clarified in the methods and figure legends. Please state how many animals were excluded for lack of cerebral blood flow reduction for rigor. The methods section only mentions use of young mice while the text describes aged mice (unclear if both sexes were used in the aged cohort and this should be added to the text, just for clarity, it would be a lot to include a full cohort of females (which may have a different microglial transcriptome) or aged mice for all study endpoints, although the transcriptomic and all the subsequent data is exclusively young males which should be clearly stated). It is noted (*and typo line 123 starting sentence with And), "And we observed a similar effect, although more pronounced, in older animals (20month old), both in Neuroscore and in the reduction of synapses. However, this effect remains

transient (Fig EV3A-C)". Was any connectivity done in the subacute stages in older animals? If not that should just be mentioned.

Was there a difference in glutamatergic synapses (lower) and sphericity (higher) just based on aging (young sham and aged sham)? It seems so based on the scales in 3b and 2c, and might be worth a mention, although if they were done in separate cohorts they can't be directly compared, and this was a small part of the paper anyway. The methods section still needs some clarifications (adding aged mice details) and as the mice seem to be housed or from different labs (note "and WT-littermates control were bred by Prof. D. Stegner at the Institute for Experimental Biomedicine, University of Würzburg"). Can you clarify if the WT controls were just bred and then transferred and then housed at the site that did the TIA surgeries? How long were they acclimated? Were all the surgeries done at the same site? It seems like some of the studies (PET vs 2P vs behavior) were done at different sites, which just requires some clarification. This is important (1-2 sentences) for a "model" paper to ensure rigor, although the comparison is really between TIA and sham, so as long as that group was in a consistent setting, it is less of a concern).

Please correct Fig 1 legend to include the females as shown in the figure (with the n, which was low but OK as the same thing was seen with a n of 5 and OK for an exploratory paper, but the legend should be clarified if the subsequent panels, BBB etc. are all male which I think they are).

It is quite surprising that no changes were seen with BCCAO (a new addition to the paper). Were ATP levels assessed in that model? Although it is interesting that they added that, it does raise some additional questions.

The limitation of the proteomics and the cellular source of the ATP have now been discussed.

The authors addressed the remaining editorial issues.

13th Feb 2026

Dear Arthur,

We are pleased to inform you that your manuscript is accepted for publication and is now being sent to our publisher to be included in the next available issue of EMBO Molecular Medicine.

You may qualify for financial assistance for your publication charges - either via a Springer Nature fully open access agreement or an EMBO initiative. Check your eligibility: <https://link.springer.com/journal/44321/how-to-publish-with-us>

Kind regards ,
Jingyi

Jingyi Hou
Senior Editor
EMBO Molecular Medicine

>>> Please note that it is EMBO Molecular Medicine policy for the transcript of the editorial process (containing referee reports and your response letter) to be published as an online supplement to each paper. If you do NOT want this, you will need to inform the Editorial Office via email immediately. More information is available here: <https://link.springer.com/partners/embo-press/editorial-policies#Peer%20review>